# SpeedVFI: One-step Diffusion for Efficient Video Frame Interpolation

**Ganggui Ding** [* 1]  **Xiaogang Xu** [* 2]  **Hao Chen** [1]  **Chunhua Shen** [1]

## Abstract

Generative video diffusion models have shown strong robustness to large motion and occlusions for video frame interpolation (VFI). However, their inference efficiency lags significantly behind learning-based methods due to the structural redundancy of pairwise inference and the procedural latency of multi-step iterative denoising. To address these limitations, we propose SpeedVFI, a task-specific one-step diffusion formulation that recasts generative VFI as unified sequence interpolation. SpeedVFI achieves dual efficiency improvements by interpolating the entire video sequence in a single forward pass to eliminate pairwise overhead, and by distilling the generation trajectory into a one-step denoising process to bypass iterative latency. To make this formulation effective for VFI, we introduce temporal RoPE alignment for temporally consistent conditioning and noise-centric partial attention to reduce computational overhead while preserving global context. Extensive experiments demonstrate that SpeedVFI accelerates diffusion-based VFI by orders of magnitude while maintaining competitive quantitative and visual quality.

## 1. Introduction

Generative models (Ho et al., 2020; Lipman et al., 2022) have recently become competitive backbones for image and video synthesis tasks (Wan et al., 2025; Hong et al., 2023; Ding et al., 2024; Yang et al., 2023). This trend has also reached video frame interpolation (VFI): compared with conventional feed-forward interpolators (Guo et al., 2024; Li et al., 2023; Reda et al., 2022; Seo et al., 2025; Ding et al., 2026), generative VFI models (Feng et al., 2024; Danier et al., 2024; Chen et al., 2025; Wang et al., 2025b) can better handle cases where the correct intermediate content is underdetermined, e.g., large motion, occlusions, or ambiguous motion boundaries.

However, the deployment of these models often remains misaligned with the practical use case of densifying sparse keyframe sequences. Given $N$ keyframes, most pipelines still reduce the problem to $N-1$ independent pairwise interpolations between adjacent frames. This design suffers from two structural inefficiencies: (i) redundant computation, since the same intermediate keyframes are repeatedly encoded and reprocessed across calls; and (ii) myopic conditioning, because each run only "sees" a local pair and cannot directly leverage global temporal context from the full keyframe sequence.

For diffusion-based VFI, both issues compound. As shown in Fig. 1(a), a pairwise diffusion pipeline performs $K$ denoising steps for each of the $N-1$ pairs, while also duplicating $N-2$ keyframes across inputs; the resulting outputs can include duplicated frames that require post-hoc fusion. Overall, this setup spends most of its compute budget on repeated work that does not increase the available information. We therefore ask a fundamental question: *Can diffusion-based VFI simultaneously process the entire keyframe sequence and generate results in one-step denoising, eliminating both pairwise redundancy and iterative latency?*

To answer this, we propose **SpeedVFI**, a task-specific formulation for efficient generative VFI. Rather than applying a diffusion model repeatedly to adjacent frame pairs, SpeedVFI recasts interpolation as unified one-step sequence generation: it processes all keyframes jointly, predicts the complete interpolated sequence in a single forward pass, and collapses the denoising trajectory into one step via a specialized distillation strategy. This formulation targets the main practical bottleneck of generative VFI: making diffusion-based interpolation efficient enough to be competitive with feed-forward systems while retaining the generative prior that is useful for ambiguous motion and occlusion. To support this high-efficiency architecture, we introduce two essential components. First, departing from simple concatenation which disrupts temporal continuity, we propose temporal RoPE alignment to ensure positional consistency between sparse observed keyframes and dense unobserved latents. Second, to mitigate the quadratic complexity of

---

[*]Equal contribution  [1]Zhejiang University, State Key Lab of CAD & CG, Hangzhou, China  [2]Zhejiang University, Hangzhou, China. Correspondence to: Chunhua Shen <Chunhua@icloud.com>.

*Proceedings of the 43$^{rd}$ International Conference on Machine Learning*, Seoul, South Korea. PMLR 306, 2026. Copyright 2026 by the author(s).

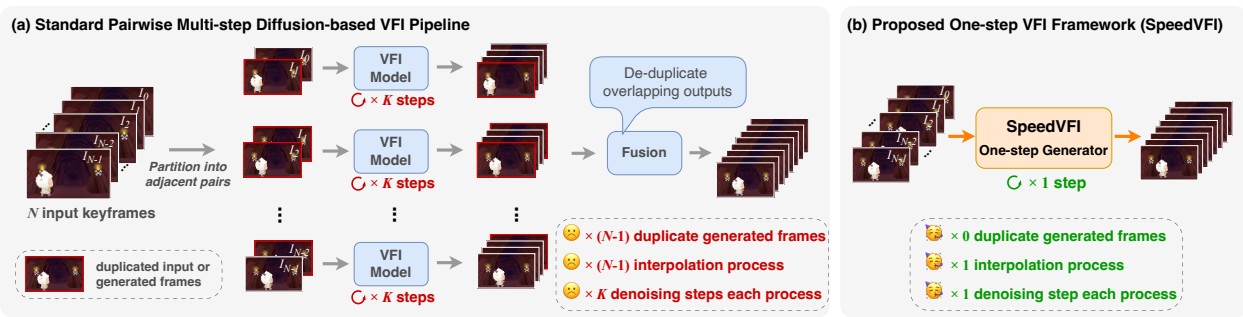

Figure 1. **Motivation.** (a) Conventional diffusion-based interpolation pipelines operate in a pairwise manner. For $N$ input keyframes, the model must run $N-1$ separate interpolation processes, repeatedly encoding $N-2$ duplicated frames and performing $K$-step denoising in each pass. This not only incurs substantial computational overhead, but also produces $N-1$ redundant output frames that require additional fusion. (b) Our SpeedVFI framework removes these inefficiencies by treating all $N$ keyframes as a unified sequence. A single forward pass with one denoising step produces the complete interpolated video without duplicated inputs or outputs.

attention over long sequences, we design a noise-centric partial attention mechanism that exploits the informational asymmetry between fixed conditions and evolving noise, prioritizing computation on the unknown frames.

Our work makes the following contributions:

- We identify redundant computation and limited temporal context as key bottlenecks of pairwise diffusion-based VFI when densifying multi-keyframe sequences.

- We propose SpeedVFI, a unified one-step sequence-interpolation formulation that removes both pairwise redundancy and iterative denoising latency in generative VFI.

- We adapt one-step diffusion distillation to VFI with temporal RoPE alignment and noise-centric partial attention, enabling efficient multi-frame conditioning with global temporal awareness.

- SpeedVFI achieves orders-of-magnitude acceleration over diffusion-based VFI baselines and reaches the practical efficiency range of strong learning-based methods, while preserving the advantages of generative modeling in challenging interpolation scenarios.

## 2. Related Work

### 2.1. Video Frame Interpolation

Video frame interpolation aims to generate intermediate frames given two or more key frames. Existing methods can be broadly categorized into learning-based and diffusion-based approaches.

**Learning-based methods.** Early VFI focuses on explicit motion modeling. Kernel-based methods (Shi et al., 2021; Chen et al., 2021; Niklaus et al., 2021; Ding et al., 2022; Lee et al., 2020; Danier et al., 2022; Zhang et al., 2023a; Zhou et al., 2023; Cheng & Chen, 2021; Ding et al., 2021; Kalluri

et al., 2023) learn adaptive kernels for pixel correspondence, while flow-based approaches (Guo et al., 2024; Seo et al., 2025; Reda et al., 2022; Zhang et al., 2023b; Huang et al., 2022; Kong et al., 2022; Briedis et al., 2025; Lew et al., 2025) guide synthesis via optical flow. While efficient for simple motion, their reliance on pixel-level correspondence fails under large displacements or occlusions, causing artifacts like ghosting. This limitation motivates exploring generative priors via diffusion models.

**Diffusion-based methods.** Recent video generation progress (Liu et al., 2024b; Blattmann et al., 2023; Wang et al., 2023; Kong et al., 2024; Wan et al., 2025; Hong et al., 2023; Yang et al., 2025b; Team, 2024) inspires diffusion-based VFI, offering strong priors for occlusions. Early works like LDMVFI (Danier et al., 2024) adapt latent diffusion for robustness. DynamicCrafter (Xing et al., 2024b) fine-tunes VideoCrafter (Chen et al., 2023) for I2V, while ToonCrafter (Xing et al., 2024a) targets cartoons. Some methods (Feng et al., 2024; Wang et al., 2025b; Zhu et al., 2025) adapt SVD (Blattmann et al., 2023) via time-reversal, fusing forward and backward generations. ViBiDSampler (Yang et al., 2025a) improves sampling, and FCVG (Zhu et al., 2025) injects priors via Control-NeXt (Peng et al., 2024). However, dual-process generation incurs high costs and potential inconsistency. Recent end-to-end frameworks (Wang et al., 2025a; Zhang et al., 2025; Wan et al., 2025) enable direct generation but remain slow due to iterative denoising, hindering real-time usage.

### 2.2. Diffusion Acceleration

Accelerating diffusion models primarily follows two paths: reducing sampling steps via distillation and improving per-step efficiency via architectural optimization.

**Step Reduction via Distillation.** Progressive distillation (Salimans & Ho, 2022) iteratively halves sampling steps by matching teacher trajectories, but typically requires

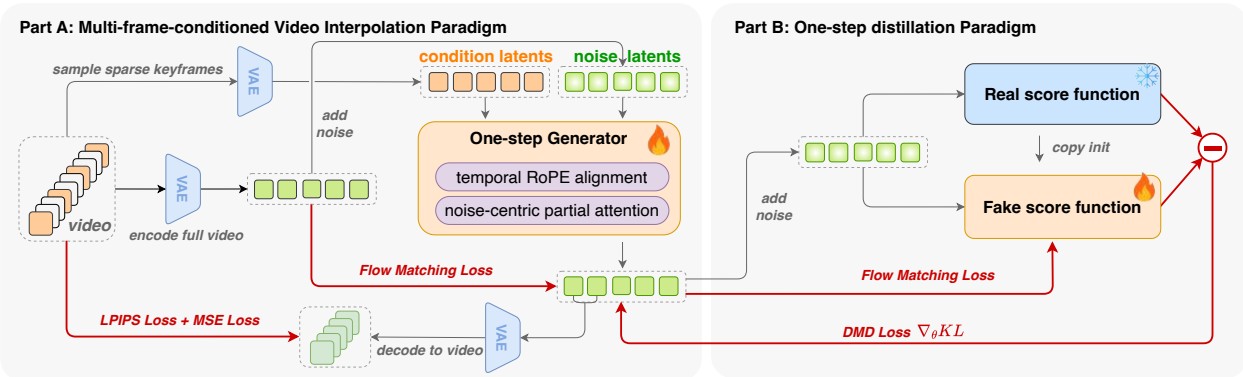

*Figure 2.* **Overview of the SpeedVFI training pipeline. Part A:** We encode keyframes into condition latents, concatenate them with noise latents, and feed the combined sequence into our one-step generator equipped with temporal RoPE alignment and noise-centric partial attention. We incorporate a Flow Matching loss and a pixel loss during training. **Part B:** We train our one-step generator with the DMD framework.

multiple distillation stages to approach very few sampling steps. Consistency-based methods (Song et al., 2023; Luo et al., 2023) learn self-consistent mappings along the sampling trajectory and are effective one- or few-step alternatives. Distribution matching distillation (DMD) (Yin et al., 2024b) and its improved variant DMD2 (Yin et al., 2024a) instead optimize the student distribution toward the teacher distribution, making them a suitable starting point for true one-step generation. Our goal is not to introduce a new generic distillation objective; rather, we adapt this one-step distribution-matching recipe to VFI, where additional motion and reconstruction constraints are needed to preserve temporal fidelity. While these acceleration methods are effective for images, scaling one-step distillation to video remains challenging due to the significantly higher training costs and spatio-temporal complexity.

**Per-Step Efficiency.** Complementary to step reduction, this line of work reduces the computational cost of each denoising step. Techniques include token merging (Bolya et al., 2022; Li et al., 2024b) to reduce sequence length, and feature caching strategies (Ma et al., 2024; Liu et al., 2024a) that reuse features across adjacent timesteps to skip redundant computations. Others propose efficient attention mechanisms such as pyramid attention broadcasting (Zhao et al., 2024) or distributed parallel inference (Li et al., 2024a) to mitigate the quadratic cost of attention. Despite reducing latency, these methods often require integration with few-step samplers to achieve real-time performance.

## 3. Method

### 3.1. Preliminary

**Flow Matching (FM) Model** learns a continuous mapping from a simple noise distribution $p_{\text{noise}}$ to the target data distribution $p_{\text{data}}$ by modeling a velocity field that governs the transformation trajectory. Unlike diffusion models that

rely on stochastic denoising processes, FM defines a deterministic flow where each intermediate state $\mathbf{z}_t$ evolves under a time-dependent velocity field $\mathbf{v}_\theta(\mathbf{z}_t, t)$, parameterized by a neural network with learnable parameters $\theta$. Given a time variable $t \in [0, 1]$, we define a linear interpolation path between clean and noisy samples:

$$\mathbf{z}_t = (1 - t)\mathbf{z}_0 + t\mathbf{z}_1, \quad \mathbf{z}_0 \sim p_{\text{data}}, \ \mathbf{z}_1 \sim p_{\text{noise}}. \quad (1)$$

The corresponding target velocity is $\mathbf{v}_t^\star = \partial \mathbf{z}_t / \partial t$ under this path. The training objective minimizes the mean squared error between the predicted and target velocities:

$$\mathcal{L}_{\text{flow}} = \mathbb{E}_{\mathbf{z}, \, t \sim \mathcal{U}(0,1)} \left\| \mathbf{v}_t^\star - \mathbf{v}_\theta(\mathbf{z}_t, t) \right\|_2^2. \quad (2)$$

After training, realistic samples can be generated by following the learned continuous flow from the noise prior ($t = 1$) toward the data manifold ($t = 0$).

**Distribution Matching Distillation (DMD)** (Yin et al., 2024b) aims to distill a multi-step diffusion process into a few-step or even one-step generator by explicitly minimizing the distributional discrepancy between the teacher diffusion model and the student generator. Instead of imitating denoising trajectories, DMD formulates the distillation objective as minimizing the Kullback–Leibler (KL) divergence between the generator-induced distribution $p_{\text{fake}}$ and the target distribution $p_{\text{real}}$, as $D_{\text{KL}}(p_{\text{fake}} \| p_{\text{real}})$. In practice, this divergence is optimized through score matching, where two score functions are introduced: a *real* score network $s_{\text{real}}$ that estimates the gradient of the log-density of the teacher diffusion model, and a *fake* score network $s_{\text{fake}}$ that estimates the score of the student generator's distribution. At each training step, a noisy version of the generator output $x_t = \alpha_t G_\theta(z) + \sigma_t \epsilon, \ \epsilon \sim \mathcal{N}(0, I)$ is fed into both $s_{\text{real}}$ and $s_{\text{fake}}$, and the generator parameters $\theta$ are updated to minimize the difference between their score predictions:

$$\nabla_\theta D_{\text{KL}} \simeq \mathbb{E}_{z, t, x_t} \left[ \left( s_{\text{fake}}(x_t, t) - s_{\text{real}}(x_t, t) \right) \frac{\partial G_\theta(z)}{\partial \theta} \right]. \quad (3)$$

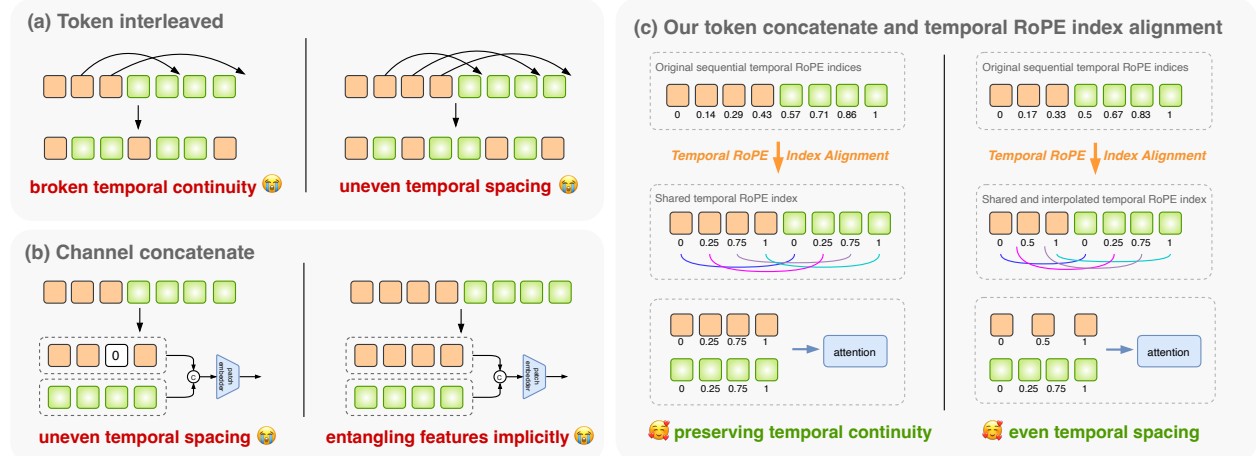

*Figure 3.* **Temporal RoPE Index Alignment.** (a) Token interleaving breaks noise continuity and causes uneven spacing. (b) Channel concatenation entangles condition and noise features. (c) Our method aligns their temporal RoPE indices via interpolation, achieving consistent spacing and continuous temporal semantics.

Here, $s_{\text{real}}$ and $s_{\text{fake}}$ represent the estimated score functions at time $t$, and $G_\theta$ denotes the student generator parameterized by $\theta$.

### 3.2. One-step Distillation Pipeline

We train our one-step generator upon the DMD (Yin et al., 2024b;a) framework. DMD aligns the output distribution of a student model with that of a multi-step teacher diffusion model through distribution matching, serving as an implicit distillation mechanism. While effective for general image or video generation, such distribution-level alignment alone is insufficient for video interpolation, where the generated intermediate frames must be both semantically consistent with neighboring keyframes and temporally smooth in motion.

To address this limitation, we enhance the original DMD objective with multi-level constraints on the one-step generator $G_\theta$, as shown in Fig. 2. In the latent space, we introduce a flow matching loss $\mathcal{L}_{\text{flow}}$ to explicitly enforce motion continuity between frames. In the pixel space, we incorporate a pixel loss $\mathcal{L}_{\text{pixel}}$ including perceptual (LPIPS) and reconstruction (MSE) losses to promote structural fidelity and perceptual realism. We denote the predicted latent velocity as $v_{\text{pred}}$, and use $v_{\text{gt}}$ and $v_{\text{fake}}$ for the ground-truth and generator-distribution velocity targets, respectively. The overall objective for $G_\theta$ is:

$$\begin{aligned} \mathcal{L}_G = \mathcal{L}_{\text{DMD}} + \mathcal{L}_{\text{flow}}(v_{\text{pred}}, v_{\text{gt}}) \\ + \mathcal{L}_{\text{flow}}(v_{\text{pred}}, v_{\text{fake}}) + \mathcal{L}_{\text{pixel}}(x_{\text{pred}}, x_{\text{gt}}), \end{aligned} \quad (4)$$

where $\mathcal{L}_{\text{DMD}}$ enforces consistency with the pretrained multi-step diffusion model, $\mathcal{L}_{\text{flow}}(v_{\text{pred}}, v_{\text{gt}})$ provides the main VFI-specific supervision by matching the predicted latent velocity to the ground-truth interpolation trajectory, $\mathcal{L}_{\text{flow}}(v_{\text{pred}}, v_{\text{fake}})$ regularizes the predicted velocity toward

the generator-induced trajectory used during DMD-style training, and $\mathcal{L}_{\text{pixel}}$ encourages the predictions to align with the real data distribution in pixel space. Furthermore, following DMD2 (Yin et al., 2024a), training stability can be improved via the Two Time-Scale Update Rule (TTUR), where $s_{\text{fake}}$ is updated more frequently than $G_\theta$ to ensure that the score estimator remains close to the generator's evolving distribution.

### 3.3. Multi-frame Interpolation

**Interpolation with Multi-frame Conditioning.** Existing VFI methods, such as GIMM-VFI (Guo et al., 2024) and Wan-FLF2V (Wan et al., 2025), typically operate in a pairwise manner, where each interpolation is performed between the first and last frames of a short clip. As shown in Fig 1, for a sequence with $N$ keyframes, the model must be invoked $(N-1)$ times to obtain the fully interpolated video, and $(N-2)$ intermediate keyframes are repeatedly fed into the model. This iterative formulation is computationally inefficient, especially for diffusion-based approaches that require dozens or thousands of denoising steps for each interpolation.

In contrast, our SpeedVFI supports multi-frame conditioning and generates the entire interpolated sequence in a *single* one-step inference. Given $N$ input keyframes

$$V_{\text{cond}} = \{I^0, I^1, \ldots, I^{N-1}\}, \quad (5)$$

we first encode each frame using the VAE encoder $\mathcal{E}$ to obtain its latent representation:

$$z_{\text{cond}} = \{\mathbf{z}_c^0, \mathbf{z}_c^1, \ldots, \mathbf{z}_c^{N-1}\}, \quad \mathbf{z}_c^n = \mathcal{E}(I^n). \quad (6)$$

Given the desired interpolation ratio $R$ (e.g., $4\times$, $8\times$), we

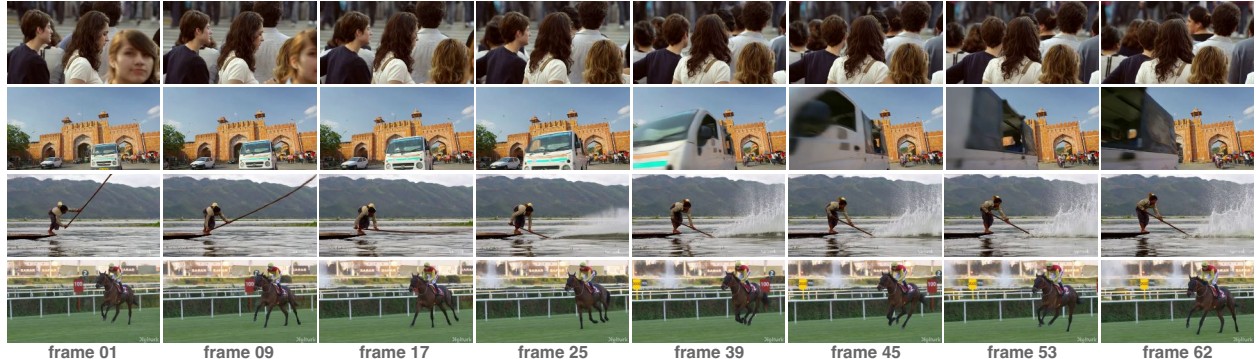

*Figure 4.* **Qualitative results.** We randomly sample intermediate frames from the 65-frame videos generated by our method.

construct a set of noise latents as

$$z_{\text{noise}} = \{\mathbf{z}_n^0, \mathbf{z}_n^1, \dots, \mathbf{z}_n^{M-1}\}, \quad M = \frac{R(N-1)}{C} + 1, \quad (7)$$

where $C$ is the temporal compression ratio of the VAE (typically $C = 4$ in CogVideoX (Yang et al., 2025b), Hunyuan-Video (Kong et al., 2024), WanVideo (Wan et al., 2025)). This formulation naturally supports any interpolation factor that is a multiple of $C$.

We then concatenate conditional latents and noise latents along the temporal dimension:

$$z = \text{concat}(z_{\text{cond}}, z_{\text{noise}}), \quad (8)$$

and apply temporal RoPE (Su et al., 2021) interpolation to align positional embeddings between the two latent groups, allowing noise latents to inherit both semantic and motion boundary cues.

The concatenated latent sequence is fed into the one-step generator $G_\theta$ to predict the clean latent: $z_0 = G_\theta(z)$. Afterward, we discard the conditional portion of $z_0$ and decode the remaining latent sequence via the VAE decoder $\mathcal{D}$ to obtain the final interpolated video:

$$V_{\text{final}} = \mathcal{D}(z_0) = \{I^0, I^1, \dots, I^{L-1}\}, \quad L = R(N-1)+1. \quad (9)$$

Through this design, SpeedVFI achieves one-step multi-frame interpolation, substantially reducing computational cost and greatly improving inference efficiency.

**Temporal RoPE Alignment.** Directly concatenating the conditional latents $z_{\text{cond}}$ and the noise latents $z_{\text{noise}}$ along the temporal dimension introduces a temporal coordinate mismatch. The model inherently interprets the concatenated sequence as physically adjacent (e.g., $t, t + 1, \dots$), whereas the noise latents should occupy continuous physical positions within the span of the condition frames. This biases the attention mechanism to treat the task as extrapolation rather than interpolation.

A naïve solution is to interleave the two latent sequences in time (Fig. 3 (a)). Interleaving preserves the chronological order of physical frames, but it breaks the continuous token trajectory formed by target noise latents: adjacent target positions become separated by condition tokens, so their token-level spacing no longer matches their physical temporal spacing, especially when the sequence lengths differ. Channel-wise concatenation (Fig. 3 (b)) implicitly entangles the two types of features within the patch embedding layer and still fails to provide a coherent temporal ordering.

To resolve this, we propose temporal RoPE alignment, which maps both conditional and noise latents onto a unified continuous temporal coordinate system before concatenation. We define the temporal coordinates for the $N$ condition frames as normalized indices:

$$t_{\text{cond}}^{(n)} = \frac{n}{N-1}, \quad n \in \{0, \dots, N-1\}. \quad (10)$$

For the generated noise latents, instead of assigning them arbitrary sequence indices, we calculate their physical timestamps $t_{\text{noise}}$ via linear interpolation over the same condition span:

$$t_{\text{noise}}^{(i)} = \frac{i}{M-1}, \quad i \in \{0, \dots, M-1\}, \quad (11)$$

where $M$ is the number of target noise latents. Crucially, we apply Rotary Positional Embedding (RoPE) based on these aligned physical coordinates rather than the token indices in the concatenated tensor:

$$\mathbf{R}(z, t) = z \cdot \mathcal{R}_\Theta(t), \quad t \in t_{\text{cond}} \cup t_{\text{noise}}, \quad (12)$$

where $\mathcal{R}_\Theta(t)$ denotes the rotation matrix governed by frequency $\Theta$ at time $t$. By enforcing this alignment, the noise latents semantically "occupy" the correct continuous positions between keyframes in the attention space, enabling the model to strictly adhere to the temporal geometry of the interpolation task.

**Condition–Noise Modulation** In original DiT blocks, the modulation parameters (scale, shift, gate) are computed from the sampling timestep $t$, meaning that all tokens are modulated according to the same denoising stage, as shown

in Fig. 5 (a). However, video interpolation naturally contains two distinct types of latents: (1) condition latents, which already lie near the data manifold and should remain stable, and (2) noise latents, which must undergo a strong transformation toward clean video content.

To better reflect this asymmetry, we assign a fixed $t = 0$ to condition latents and $t = 1$ to noise latents tailored for our one-step distillation framework. This gives condition latents a near-identity modulation while allowing noise latents to receive maximal modulation, improving stability and controllability when jointly processing multi-frame inputs.

**Noise-centric Partial Attention.** We further introduce a noise-centric partial attention mechanism to reduce attention computation while preserving temporal global context. Specifically, only the queries corresponding to noise latents participate in self- and cross-attention:

$$A_{\text{noise}} = \mathcal{S}\left( \frac{Q_{\text{noise}}(K_{\text{noise}} \| K_{\text{cond}})^{\top}}{\sqrt{d_k}} \right) (V_{\text{noise}} \| V_{\text{cond}}). \quad (13)$$

$\mathcal{S}$ represent the operation of Softmax, $\|$ denotes concatenation. In this formulation, noise latents can still attend to both noise and condition context, while condition latents bypass self-attention entirely. The complexity is reduced from $\mathcal{O}((N + M)^2)$ to $\mathcal{O}(M(N + M))$, where $N$ and $M$ denote the numbers of condition and noise latents respectively.

We also apply the same principle to cross-attention layers: only noise queries interact with the text-conditioning keys and values, while condition latents skip cross-attention. Combined with dual-timestep modulation, partial self- and cross-attention increase the generation speed, yet maintain full access to conditional supervision.

## 4. Experiments

### 4.1. Implementation Details.

We train SpeedVFI on the OpenVid-HD (Nan et al., 2024) dataset, which contains a large collection of high-quality long-range videos. Each training clip is resized to $832 \times 448$, and we extract the first 65 frames. We train our model based on the DMD (Yin et al., 2024a) framework. We initialize the $s_{\text{real}}$, $s_{\text{fake}}$, and $G_\theta$ with the pretrained Wan-TI2V-5B (Wan et al., 2025) model. During training, we freeze $s_{\text{real}}$ and jointly optimize $s_{\text{fake}}$ and $G_\theta$ using TTUR with an interval of 5, meaning that $s_{\text{fake}}$ is updated five times for every update of $G_\theta$. We train on 8 A100 GPUs for 10K generator steps, using a total batch size of 8. Training takes approximately 6 days. We adopt AdamW with a learning rate of $2 \times 10^{-6}$, and all loss terms in Eq. 4 are assigned equal weights with 1. We use gradient checkpointing for the generator, score networks, and FlashAttention (Dao et al., 2022) for transformer attention modules. Because

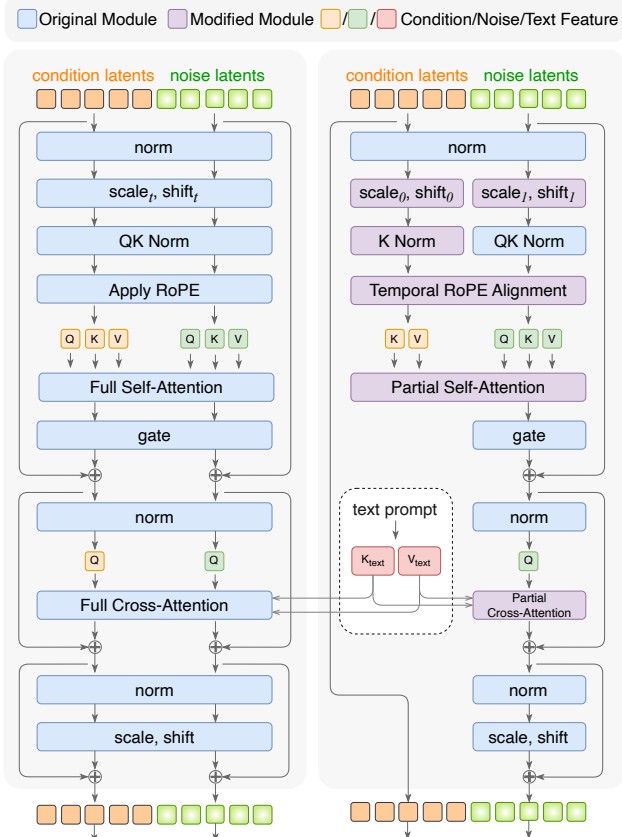

(a) Original DiT Block with Full Attention   (b) Our DiT Block with Partial Attention

*Figure 5.* **Comparison between full attention and our partial attention DiT block.** (a) Standard DiT applies timestep-dependent modulation and full self/cross-attention uniformly to all latents. (b) Our design assigns fixed timesteps ($t$=0 for condition, $t$=1 for noise), and performs partial self- and cross-attention on noise latents only, yielding significant computational savings while retaining access to all conditioning information.

full-sequence VAE decoding for pixel-space supervision remains memory intensive under this high-resolution long-video setting, we decode the first two generated latents to compute $\mathcal{L}_{\text{pixel}}$.

### 4.2. Comparisons

**Baseline Methods.** Since SpeedVFI is designed for efficient diffusion-based interpolation, we compare it against both learning-based and diffusion-based VFI approaches. The learning-based baselines include AMT (Li et al., 2023), BiM-VFI (Seo et al., 2025), and GIMM-VFI (Guo et al., 2024), which represent the state-of-the-art (SOTA) in fast interpolation. We additionally include RIFE (Huang et al., 2022) and MoMo (Lew et al., 2025) in the cost-performance analysis and appendix comparisons, as they are strong efficient flow-based systems; MoMo uses diffusion to estimate bidirectional flow but still synthesizes frames through

flow-guided warping. The diffusion-based baselines include ViBiDSampler (Yang et al., 2025a), FCVG (Zhu et al., 2025), and Framer (Wang et al., 2025a). **Evaluation Protocol.** All methods take 17 keyframes as input and perform $4\times$ interpolation to produce a 65-frame output sequence. Following Framer, we adopt PSNR and SSIM to measure reconstruction quality, LPIPS to assess perceptual similarity, and FID/FVD to evaluate image-to-video realism. **Evaluation Data.** For comprehensive benchmarking, we use two test sets: one containing 24 long videos selected from the DAVIS (Pont-Tuset et al., 2017) dataset, and another which we term DiversityVideo, is a specialized collection of 24 high-quality videos collected from BVI-DVC (Ma et al., 2021) and SNU-FILM (Choi et al., 2020) datasets. Across these 48 videos, each is temporally trimmed to 65 frames, from which 17 keyframes are uniformly sampled for interpolation. This results in a substantial total of $16 \times 48 = 768$ input pairs. All methods are evaluated at their native inference resolution to ensure fair comparison.

**Qualitative Results** As shown in Fig. 4, SpeedVFI successfully interpolates 65 frames from sparse keyframes, maintaining stable performance in high-motion scenes like equestrianism, where fine details such as horse leg movement and splashing water are accurately synthesized. Furthermore, comparisons with state-of-the-art (SoTA) methods (Fig. 6) highlight SpeedVFI's exceptional robustness against large motion and complex details. Traditional flow-based techniques often yield artifacts or detail loss, for instance, blurring fine architectural pillars, breaking thin flamingo legs, or losing small objects like a held stick, due to occlusion and estimation errors. In contrast, our generative, one-step framework reliably preserves these challenging details, achieving superior perceptual quality and demonstrating robustness for real-world VFI applications.

**Quantitative Results** We evaluate SpeedVFI against existing methods on the DAVIS and DiversityVideo datasets. The results are summarized in Table 1. To provide a comprehensive landscape of current VFI research, we include both traditional learning-based methods and recent diffusion-based generative approaches. For clarity, learning-based methods are shown in gray as a reference; bold and underlined values indicate the best performance overall and the best results specifically within the diffusion-based category, respectively. SpeedVFI achieves performance competitive with traditional learning-based VFI methods, even surpassing them in several key metrics such as FVD. Crucially, our method significantly outperforms existing diffusion-based VFI approaches, demonstrating that our one-step distillation and multi-frame conditioning strategy mitigates the efficiency-performance trade-off typically associated with diffusion models. Appendix D reports detailed RIFE/MoMo and ViBiDSampler-Wan comparisons, Appendix E reports

$8\times$ generalization plus runtime and memory scaling.

**Inference Speed.** As shown in Table 2, we evaluate the inference latency on an NVIDIA A100 GPU at a resolution of $832 \times 448$. While conventional learning-based models are naturally efficient, existing diffusion-based methods are typically plagued by the multi-step sampling bottleneck (e.g., 30–50 NFE). By virtue of our one-step distillation and parallel multi-frame conditioning, SpeedVFI effectively closes this efficiency gap. Our method achieves an inference time of 9.98s for generating 65 frames, representing a $15\times$ to $157\times$ speedup over current diffusion-based competitors like Framer and FCVG. Remarkably, our model even slightly outperforms the SoTA conventional method BiM-VFI (14.16s), demonstrating that generative VFI can achieve real-time efficiency parity with learning-based models.

**Cost-performance Analysis.** Latency alone can obscure the total task cost, because pairwise methods require repeated recursive interpolation and diffusion methods require multiple function evaluations. Fig. 7 therefore visualizes task-level TFLOPs for the complete $17 \rightarrow 65$, $4\times$, $832 \times 448$ setting, with the exact metadata and numeric values reported in Appendix B. We include RIFE and MoMo in this cost-performance view because they are representative efficient flow-based systems and provide useful low-cost reference points for interpreting the efficiency frontier; to keep the main quantitative and qualitative comparisons compact, their detailed results are reported in Appendix D. SpeedVFI is not designed to minimize FLOPs against every lightweight conventional model; rather, it closes the practical efficiency gap for generative VFI. Compared with diffusion-based baselines, SpeedVFI achieves the highest PSNR while using substantially fewer TFLOPs and lower latency. Compared with efficient flow-based methods such as RIFE and MoMo, SpeedVFI uses higher compute but provides stronger generative robustness in challenging occlusion and large-motion cases, as shown in Appendix D.

### 4.3. Ablation Study

**Effect of Training Losses.** To adapt the one-step generator for video interpolation, we extend the original DMD distillation framework based on $\nabla_\theta D_{\mathrm{KL}}$ by introducing two additional losses $\mathcal{L}_{\mathrm{flow}}$ and $\mathcal{L}_{\mathrm{pixel}}$ to enhance temporal coherence and reconstruction fidelity (Sec. 3.2). We conduct ablation experiments to evaluate their effects. As shown in Table 3, the Flow Matching loss yields a substantial performance gain, while the pixel loss provides further improvement. Notably, the flow-based supervision plays a pivotal role in preserving temporal consistency for video interpolation. Qualitative results in Fig. 8 further confirm that the flow loss is essential for maintaining identity coherence and stable temporal dynamics during video synthesis. Ap-

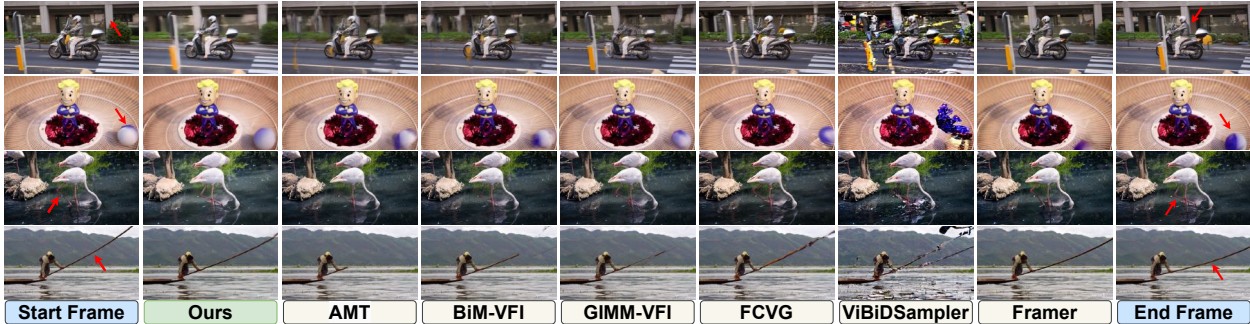

*Figure 6.* **Qualitative Comparison.** Visual comparison between SpeedVFI and other state-of-the-art methods on challenging sequences. Red arrows highlight regions where our method better preserves structural integrity and fine-grained textures. Please zoom in for better visualization.

*Table 1.* Quantitative comparison of our method with SoTA learning-based and diffusion-based methods on the DAVIS and Diversity datasets. Our approach demonstrates competitive performance with leading learning-based methods and outperforms diffusion-based methods across multiple evaluation metrics. **Bold** and underline indicate the best results overall and among diffusion-based methods, respectively. Learning-based methods are shown in gray for reference.

| Methods | DAVIS | | | | | DiversityVideo | | | | |
|---|---|---|---|---|---|---|---|---|---|---|
| *Learning-based* | PSNR ↑ | SSIM ↑ | LPIPS ↓ | FID ↓ | FVD ↓ | PSNR ↑ | SSIM ↑ | LPIPS ↓ | FID ↓ | FVD ↓ |
| AMT-S (Li et al., 2023) | 23.56 | 0.77 | 0.183 | 41.50 | 640.89 | 30.13 | 0.94 | 0.062 | 18.99 | 555.96 |
| AMT-L (Li et al., 2023) | 23.64 | 0.77 | 0.184 | 43.36 | 633.72 | 30.21 | 0.94 | 0.062 | 19.11 | 558.84 |
| BiM-VFI (Seo et al., 2025) | 26.90 | 0.84 | 0.114 | 26.78 | 214.66 | 34.65 | 0.98 | 0.041 | 13.39 | 136.87 |
| GIMM-VFI (Guo et al., 2024) | 27.10 | 0.86 | 0.112 | 28.10 | 210.45 | 34.86 | 0.98 | 0.035 | 12.44 | 132.63 |
| *Diffusion-based* | PSNR ↑ | SSIM ↑ | LPIPS ↓ | FID ↓ | FVD ↓ | PSNR ↑ | SSIM ↑ | LPIPS ↓ | FID ↓ | FVD ↓ |
| ViBiDSampler (Yang et al., 2025a) | 19.89 | 0.63 | 0.246 | 29.56 | 940.89 | 24.43 | 0.84 | 0.143 | 26.83 | 1558.1 |
| FCVG (Zhu et al., 2025) | 22.30 | 0.73 | 0.191 | 28.60 | 430.23 | 21.94 | 0.73 | 0.149 | 18.30 | 373.76 |
| Framer (Wang et al., 2025a) | 22.79 | 0.76 | 0.190 | 34.11 | 338.51 | 26.22 | 0.89 | 0.113 | 29.96 | 401.89 |
| **SpeedVFI** (this work) | **27.12** | **0.87** | 0.123 | 27.01 | **205.69** | 34.71 | 0.97 | 0.038 | 14.50 | **116.68** |

*Table 2.* Inference speed comparison on A100 GPU. Each method interpolates 17 key frames to 65 frames at $832 \times 448$ resolution. NFE denotes the Number of Function Evaluations. "Conv." denotes conventional learning-based methods, and "Diff." denotes diffusion-based VFI methods.

| Method | Venue | Category | NFE | Time (s)↓ |
|---|---|---|---|---|
| AMT-S (Li et al., 2023) | CVPR'23 | Conv. | - | 10.32 |
| AMT-L (Li et al., 2023) | CVPR'23 | Conv. | - | 10.56 |
| GIMM-VFI (Guo et al., 2024) | NeurIPS'24 | Conv. | - | 10.54 |
| BiM-VFI (Seo et al., 2025) | CVPR'25 | Conv. | - | 14.16 |
| ViBiDSampler (Yang et al., 2025a) | ICLR'25 | Diff. | 50 | 371.40 |
| FCVG (Zhu et al., 2025) | CVPR'25 | Diff. | 50 | 1570.18 |
| Framer (Wang et al., 2025a) | ICLR'25 | Diff. | 30 | 150.69 |
| **SpeedVFI w/ Full Attention** | this work | Diff. | 1 | 10.72 |
| **SpeedVFI w/ Partial Attention** | this work | Diff. | 1 | **9.98** |

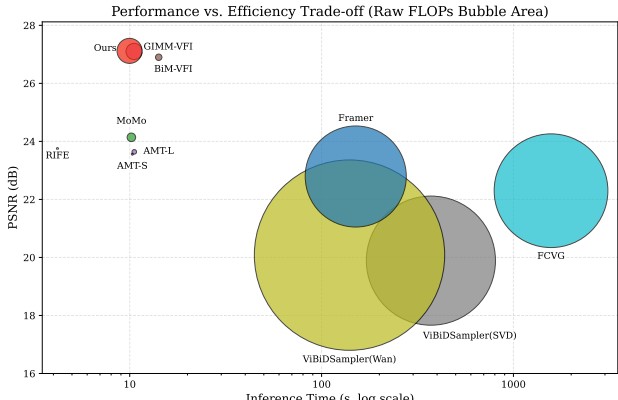

*Figure 7.* **Task-level cost-performance comparison on DAVIS.** The x-axis shows inference time, the y-axis shows PSNR, and bubble area is proportional to task-level TFLOPs under the complete $17 \rightarrow 65$, $4\times$, $832 \times 448$ setting.

pendix F provides a detailed partial-vs-full pixel-supervision ablation.

**Token Concatenation Strategies.** As shown in Table 4, the interleaved concatenation produces visually plausible intermediate frames but suffers from noticeable blurriness in both objects and backgrounds. We attribute this degradation to disrupted temporal continuity in the noise latents, which weakens temporal interactions during denoising. Sequential concatenation mitigates this issue and improves local temporal smoothness, yet it still struggles to maintain coherent global dynamics between key frames. In contrast, our sequential strategy with RoPE alignment achieves the best quantitative performance, demonstrating the effectiveness of temporal positional alignment in enhancing temporal consistency and overall video fidelity.

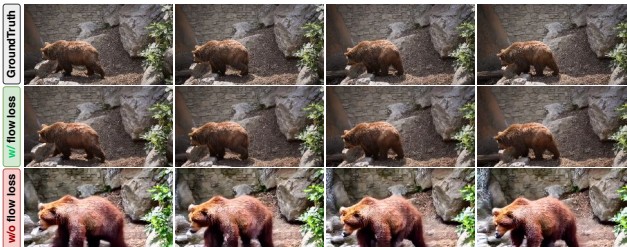

*Figure 8.* Removing $\mathcal{L}_{\text{flow}}$ leads to semantically consistent yet identity-misaligned frames, where the generated subjects exhibit degraded fidelity compared to the ground truth.

*Table 3.* Ablation on different training losses.

| $\mathcal{L}_{\text{DMD}}$ | $\mathcal{L}_{\text{flow}}$ | $\mathcal{L}_{\text{pixel}}$ | PSNR↑ | SSIM↑ | LPIPS↓ | FID↓ | FVD↓ |
|---|---|---|---|---|---|---|---|
| ✓ | | | 12.60 | 0.38 | 0.3800 | 93.40 | 1627.51 |
| ✓ | ✓ | | 20.05 | 0.65 | 0.1444 | 47.12 | 417.85 |
| ✓ | ✓ | ✓ | **20.11** | **0.65** | **0.1377** | **41.42** | **370.77** |

**Effect of TTUR Interval.** DMD2(Yin et al., 2024a) introduces the Two Time-scale Update Rule (TTUR) during distillation, where the fake score function $s_{\text{fake}}$ is updated more frequently while the one-step generator $G_\theta$ is updated at a slower pace, stabilizing the distillation dynamics. Following this strategy, we conduct an ablation study comparing two update intervals, *interval=1* and *interval=5*. As shown in Table 5, our method maintains consistent performance under both settings, with a slight improvement at *interval=5*. We attribute this robustness to the incorporated flow matching loss, which accelerates the convergence of the $G_\theta$ toward the target distribution of $s_{\text{real}}$. Therefore, we adopt *interval=5* in practice to maximize the stability and efficiency of the DMD-based training framework.

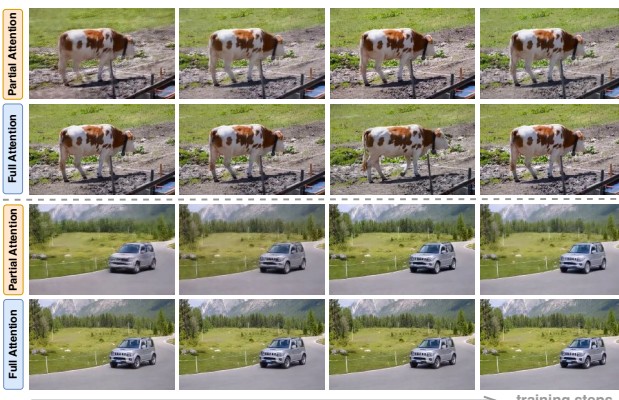

*Figure 9.* **Comparison between Full Attention and our proposed Partial Attention.** Partial Attention shows slightly lower fidelity at the early training stage but quickly converges to match Full Attention, achieving comparable reconstruction quality with higher inference efficiency.

**Full Attention vs. Partial Attention.** We further analyze the difference between our proposed Partial Attention and

*Table 4.* Comparison of different token concatenation strategies. "Interleaved" denotes temporal interleaved concatenation; "Sequential" denotes sequential concatenation without temporal RoPE alignment; "Seq. w/ RoPE" denotes sequential concatenation with temporal RoPE alignment (ours).

| Method | PSNR↑ | SSIM↑ | LPIPS↓ | FID↓ | FVD↓ |
|---|---|---|---|---|---|
| Interleaved | 19.62 | 0.63 | 0.1859 | 63.25 | 641.64 |
| Sequential | 19.94 | 0.64 | 0.1723 | 59.86 | 518.06 |
| Seq. w/ RoPE (Ours) | **20.05** | **0.65** | **0.1444** | **47.12** | **417.85** |

*Table 5.* Comparison of different update intervals in TTUR.

| interval | PSNR↑ | SSIM↑ | LPIPS↓ | FID↓ | FVD↓ |
|---|---|---|---|---|---|
| 1 | 20.05 | 0.65 | 0.1603 | 52.88 | 470.75 |
| 5 | **20.06** | **0.65** | **0.1360** | **44.19** | **396.13** |

*Table 6.* Quantitative ablation of Full and Partial Attention under the $832 \times 448$, $9 \to 33$, $4\times$ ablation protocol. DiT time and TFLOPs count only the denoising transformer.

| Attention | Steps | PSNR↑ | LPIPS↓ | FVD↓ | DiT Time (s)↓ | DiT TFLOPs↓ |
|---|---|---|---|---|---|---|
| Full | 1K | 20.11 | 0.1377 | 370.77 | 2.12 | 75.44 |
| Partial | 1K | 20.06 | 0.1480 | 382.57 | **1.44** | **53.80** |
| Partial | 2K | **20.14** | **0.1212** | **347.53** | **1.44** | **53.80** |

the original Full Attention. As shown in Fig. 9 and Table 6, Partial Attention exhibits a small optimization lag under the same 1K-step budget, but it recovers with longer optimization and reaches comparable or better reconstruction quality. This observation suggests that Partial Attention initially perturbs the model's attention prior, while the generator quickly adapts to this new inductive bias during optimization. Importantly, the efficiency gain is clearer inside the denoising transformer: in the $832 \times 448$, $9 \to 33$ ablation setting, Partial Attention reduces DiT TFLOPs from 75.44 to 53.80. Under the full $17 \to 65$ setting, transformer TFLOPs decrease from 168.54 to 127.66, while the end-to-end latency improvement in Table 2 is diluted by fixed VAE encode/decode costs. Full cross-setting results are provided in Appendix C.

## 5. Conclusion

In this work, we present SpeedVFI, a unified one-step diffusion framework for efficient generative video frame interpolation. By combining single-pass multi-frame conditioning with temporal RoPE alignment and noise-centric partial attention, SpeedVFI reduces redundant pairwise computation and removes iterative sampling latency. Extensive experiments show that SpeedVFI substantially accelerates diffusion-based VFI while preserving strong reconstruction and perceptual quality, bringing generative interpolation into the practical efficiency range of fast learning-based systems. These results suggest that task-specific formulation and system-level efficiency design are key to making generative VFI practical for long-sequence interpolation.

## Acknowledgements

This work was supported by the National Natural Science Foundation of China (No. 62576315).

## Impact Statement

This work aims to improve the efficiency of diffusion-based video frame interpolation, which can benefit applications such as video restoration, slow-motion generation, and efficient content creation. At the same time, lowering the cost of high-quality video generation and interpolation may also lower the barrier for producing manipulated media, including deepfakes, synthetic evidence, or misinformation. We therefore encourage responsible use with provenance tracking, watermarking, detection tools, and clear user consent when interpolated or generated videos are distributed.

The current system also has technical limitations. It is mainly validated with the Wan-TI2V-5B backbone, and broader cross-backbone evaluation remains future work. Under high-resolution long-sequence training, pixel-space supervision is computed only on the first two generated latents due to memory constraints, and longer videos or higher resolutions may further increase computation and latency. Challenging cases such as severe occlusion, tiny objects, fast non-rigid motion, and large viewpoint changes can still expose a sharpness-smoothness trade-off in one-step generation.

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

## A. Details of Ablation Study

For all ablation experiments, we train the models on the OpenVid-HD (Nan et al., 2024) dataset using a spatial resolution of $832 \times 448$. Each experiment takes 9 keyframes as input and performs $4\times$ interpolation to generate a 33-frame sequence. To ensure fair comparison across different configurations, the one-step generator is consistently trained for 1K steps in every ablation setting.

For quantitative and qualitative evaluation, we construct a validation set by selecting 48 videos from the DAVIS (Pont-Tuset et al., 2017) dataset, each temporally trimmed to 33 frames. All reported ablation results are obtained on this fixed validation set.

## B. Cost-performance Metadata

Table 7 reports the exact metadata used for the task-level cost-performance plot in Fig. 7. All TFLOPs are counted for the complete interpolation task rather than for a single pairwise forward or a single denoising step.

*Table 7.* Task-level cost-performance comparison on the DAVIS split. All costs are measured for the complete $832 \times 448$, $17 \rightarrow 65$, $4\times$ interpolation task, including recursive pairwise calls or all sampling steps where applicable.

| Method | Type | PSNR↑ | Time (s)↓ | TFLOPs↓ |
|---|---|---|---|---|
| RIFE (Huang et al., 2022) | Conv. | 23.75 | 4.20 | 4.54 |
| MoMo (Lew et al., 2025) | Conv. | 24.14 | 10.20 | 76.87 |
| AMT-S (Li et al., 2023) | Conv. | 23.56 | 10.32 | 4.11 |
| AMT-L (Li et al., 2023) | Conv. | 23.64 | 10.56 | 21.93 |
| BiM-VFI (Seo et al., 2025) | Conv. | 26.90 | 14.16 | 43.35 |
| GIMM-VFI (Guo et al., 2024) | Conv. | 27.10 | 10.54 | 277.36 |
| ViBiDSampler (Yang et al., 2025a) | Diff. | 19.89 | 371.40 | 17584.26 |
| ViBiDSampler-Wan | Diff. | 20.08 | 139.83 | 38213.20 |
| FCVG (Zhu et al., 2025) | Diff. | 22.30 | 1570.18 | 13637.28 |
| Framer (Wang et al., 2025a) | Diff. | 22.79 | 150.69 | 10773.10 |
| **SpeedVFI** | Diff. | **27.12** | 9.98 | 660.19 |

## C. Partial Attention Across Settings

Table 8 provides the full cross-setting quantitative ablation of Full Attention and Partial Attention. The results show that Partial Attention has a small optimization lag under the same 1K-step budget, but recovers with longer training while consistently reducing denoising-transformer cost.

*Table 8.* Full cross-setting quantitative ablation of Partial Attention. DiT time and TFLOPs count only the denoising transformer.

| Setting | Attention | Steps | PSNR↑ | LPIPS↓ | FVD↓ | DiT Time (s)↓ | DiT TFLOPs↓ |
|---|---|---|---|---|---|---|---|
| $416 \times 224$, $9 \rightarrow 33$, $4\times$ | Full | 1K | 20.25 | 0.1192 | 390.11 | 1.19 | 16.34 |
| $416 \times 224$, $9 \rightarrow 33$, $4\times$ | Partial | 1K | 20.20 | 0.1257 | 410.84 | 0.90 | 10.93 |
| $416 \times 224$, $9 \rightarrow 33$, $4\times$ | Partial | 2K | **20.48** | **0.1153** | **309.10** | 0.90 | 10.93 |
| $832 \times 448$, $9 \rightarrow 33$, $4\times$ | Full | 1K | 20.11 | 0.1377 | 370.77 | 2.12 | 75.44 |
| $832 \times 448$, $9 \rightarrow 33$, $4\times$ | Partial | 1K | 20.06 | 0.1480 | 382.57 | 1.44 | 53.80 |
| $832 \times 448$, $9 \rightarrow 33$, $4\times$ | Partial | 2K | **20.14** | **0.1212** | **347.53** | 1.44 | 53.80 |
| $832 \times 448$, $17 \rightarrow 65$, $4\times$ | Full | 1K | 18.14 | 0.4104 | 921.80 | 2.63 | 168.54 |
| $832 \times 448$, $17 \rightarrow 65$, $4\times$ | Partial | 1K | 18.02 | 0.4523 | 1002.70 | 1.78 | 127.66 |
| $832 \times 448$, $17 \rightarrow 65$, $4\times$ | Partial | 2K | **18.15** | **0.3492** | **890.31** | 1.78 | 127.66 |

## D. Additional RIFE, MoMo, and Backbone-controlled Comparisons

Table 9 adds RIFE (Huang et al., 2022), MoMo (Lew et al., 2025), and ViBiDSampler-Wan to the quantitative comparison. ViBiDSampler-Wan denotes our controlled reproduction of ViBiDSampler using the same Wan-TI2V-5B backbone as SpeedVFI, which helps separate the effect of the interpolation framework from the effect of the underlying video diffusion

*Table 9.* Additional quantitative comparisons including RIFE, MoMo, and ViBiDSampler-Wan. ViBiDSampler-Wan denotes our reproduction of ViBiDSampler with the same Wan-TI2V-5B backbone used by SpeedVFI.

| Methods | DAVIS | | | | | DiversityVideo | | | | |
|---|---|---|---|---|---|---|---|---|---|---|
| *Learning/flow-based* | PSNR ↑ | SSIM ↑ | LPIPS ↓ | FID ↓ | FVD ↓ | PSNR ↑ | SSIM ↑ | LPIPS ↓ | FID ↓ | FVD ↓ |
| AMT-S (Li et al., 2023) | 23.56 | 0.77 | 0.183 | 41.50 | 640.89 | 30.13 | 0.94 | 0.062 | 18.99 | 555.96 |
| AMT-L (Li et al., 2023) | 23.64 | 0.77 | 0.184 | 43.36 | 633.72 | 30.21 | 0.94 | 0.062 | 19.11 | 558.84 |
| BiM-VFI (Seo et al., 2025) | 26.90 | 0.84 | 0.114 | 26.78 | 214.66 | 34.65 | 0.98 | 0.041 | 13.39 | 136.87 |
| GIMM-VFI (Guo et al., 2024) | 27.10 | 0.86 | 0.112 | 28.10 | 210.45 | 34.86 | 0.98 | 0.035 | 12.44 | 132.63 |
| RIFE (Huang et al., 2022) | 23.75 | 0.79 | 0.162 | 30.15 | 554.81 | 31.24 | 0.95 | 0.055 | 18.43 | 403.21 |
| MoMo (Lew et al., 2025) | 24.14 | 0.76 | 0.168 | 32.19 | 379.16 | 32.66 | 0.95 | 0.060 | 14.65 | 209.64 |
| *Diffusion-based* | PSNR ↑ | SSIM ↑ | LPIPS ↓ | FID ↓ | FVD ↓ | PSNR ↑ | SSIM ↑ | LPIPS ↓ | FID ↓ | FVD ↓ |
| ViBiDSampler (Yang et al., 2025a) | 19.89 | 0.63 | 0.246 | 29.56 | 940.89 | 24.43 | 0.84 | 0.143 | 26.83 | 1558.10 |
| ViBiDSampler-Wan | 20.08 | 0.69 | 0.203 | 31.62 | 1022.70 | 25.02 | 0.85 | 0.127 | 27.39 | 1612.90 |
| FCVG (Zhu et al., 2025) | 22.30 | 0.73 | 0.191 | 28.60 | 430.23 | 21.94 | 0.73 | 0.149 | 18.30 | 373.76 |
| Framer (Wang et al., 2025a) | 22.79 | 0.76 | 0.190 | 34.11 | 338.51 | 26.22 | 0.89 | 0.113 | 29.96 | 401.89 |
| **SpeedVFI** (this work) | **27.12** | **0.87** | **0.123** | **27.01** | **205.69** | **34.71** | **0.97** | **0.038** | **14.50** | **116.68** |

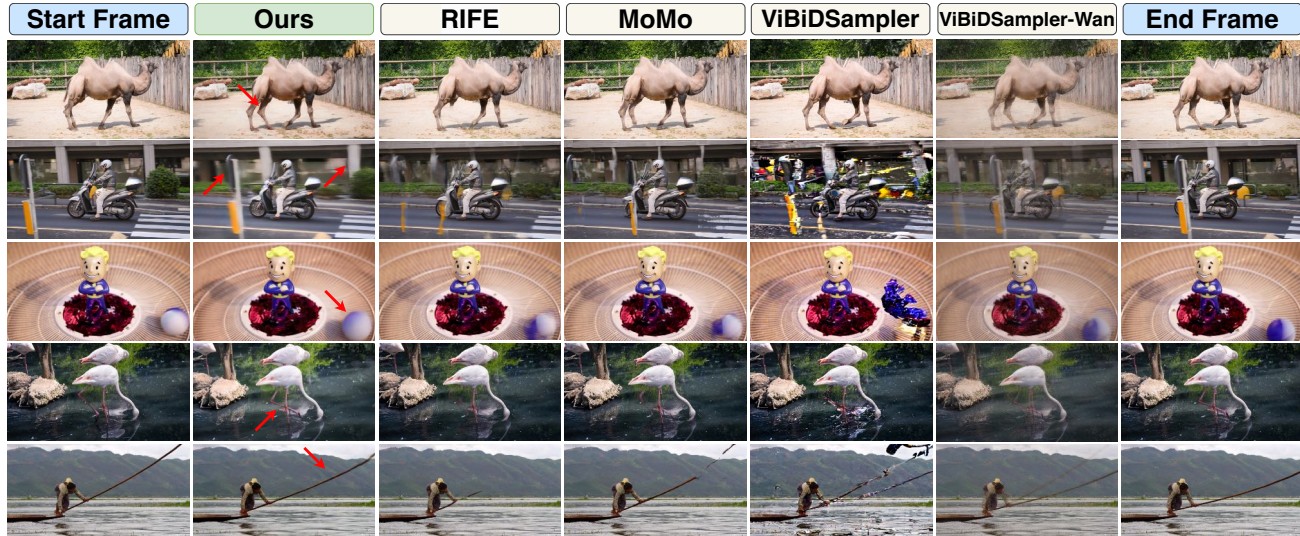

*Figure 10.* **Additional qualitative comparisons with RIFE, MoMo, ViBiDSampler, and ViBiDSampler-Wan.** SpeedVFI shows stronger robustness in challenging cases such as occlusion and fast motion. Flow-based methods can miss newly exposed structures when correspondence estimation is unreliable, while multi-step diffusion baselines may introduce temporal inconsistency or duplicated structures.

backbone. MoMo uses a diffusion model to estimate bidirectional optical flows, but its final interpolation still relies on flow-guided synthesis; therefore, we group it with flow-based interpolation systems in this comparison. For all methods, FLOPs in the cost-performance analysis are counted over the complete $17 \rightarrow 65$ interpolation process rather than a single forward call. Thus, pairwise recursive methods are charged for all interval-level calls: each adjacent keyframe interval requires three recursive $2\times$ forwards, yielding $16 \times 3 = 48$ forwards in total; diffusion-based methods are charged for all denoising evaluations needed to generate the full output sequence.

## E. Generalization, Runtime Scaling, and Memory Scaling

We further evaluate SpeedVFI beyond the main $17 \rightarrow 65$, $4\times$ setting. Table 10 reports an $8\times$ interpolation setting on DiversityVideo. Table 11 summarizes runtime scaling across different resolutions and sequence lengths; these numbers are measured on A800 and are included only as relative scaling evidence. Table 12 reports peak inference memory at $832 \times 448$, showing a modest increase across the tested range. Longer videos and higher resolutions can still increase compute and latency, and remain an important practical consideration.

*Table 10.* Generalization to $8\times$ interpolation on DiversityVideo at $832 \times 448$ resolution.

| Method | PSNR↑ | SSIM↑ | LPIPS↓ | FID↓ | FVD↓ | Time (s)↓ | TFLOPs↓ |
|---|---|---|---|---|---|---|---|
| RIFE (Huang et al., 2022) | 28.04 | 0.91 | 0.1813 | 38.90 | 355.71 | 5.28 | 5.30 |
| MoMo (Lew et al., 2025) | 29.83 | 0.91 | 0.1714 | 33.69 | 286.46 | 12.55 | 89.68 |
| ViBiDSampler-Wan | 19.79 | 0.69 | 0.3920 | 116.54 | 2769.87 | 86.93 | 26759.93 |
| Framer (Wang et al., 2025a) | 23.15 | 0.77 | 0.2398 | 34.53 | 814.63 | 92.72 | 5386.56 |
| **SpeedVFI** | **31.62** | **0.93** | **0.1146** | **27.88** | **204.14** | 10.47 | 629.75 |

*Table 11.* Runtime scaling of SpeedVFI under different interpolation settings. The numbers are measured on an NVIDIA A800 and are reported as relative scaling evidence; the main fixed-protocol runtime in Table 2 is measured on A100.

| Resolution | Condition frames | Generated frames | Time (s)↓ |
|---|---|---|---|
| $416 \times 224$ | 5 | 33 | 4.31 |
| $416 \times 224$ | 9 | 33 | 4.59 |
| $416 \times 224$ | 9 | 65 | 5.75 |
| $416 \times 224$ | 17 | 65 | 5.92 |
| $832 \times 448$ | 5 | 33 | 7.14 |
| $832 \times 448$ | 9 | 33 | 7.39 |
| $832 \times 448$ | 9 | 65 | 10.47 |
| $832 \times 448$ | 17 | 65 | 10.63 |

*Table 12.* Peak inference memory of SpeedVFI at $832 \times 448$ resolution under $4\times$ interpolation.

| Condition frames | Generated frames | Peak memory (GiB)↓ |
|---|---|---|
| 5 | 17 | 20.40 |
| 9 | 33 | 20.48 |
| 13 | 49 | 20.54 |
| 17 | 65 | 20.62 |
| 21 | 81 | 20.68 |
| 25 | 97 | 20.75 |

## F. Effect of Pixel Supervision

The main model uses partial pixel supervision because full-sequence VAE decoding for pixel-space losses is memory intensive in the high-resolution long-video training setup. Table 13 compares DMD-only training, partial/full pixel supervision, and flow-based supervision. The results indicate that pixel loss is helpful, while the flow loss provides the main global temporal and structural guidance. With flow supervision, partial pixel loss recovers most of the benefit of full pixel supervision under the practical memory budget.

*Table 13.* Ablation of partial and full pixel supervision. Partial pixel supervision decodes only the first two generated latents for $\mathcal{L}_{\text{pixel}}$.

| Variant | Losses / pixel supervision | PSNR↑ | SSIM↑ | LPIPS↓ | FID↓ | FVD↓ |
|---|---|---|---|---|---|---|
| A | DMD | 12.54 | 0.12 | 0.8217 | 402.60 | 4184.76 |
| B | DMD + partial $\mathcal{L}_{\text{pixel}}$ | 16.37 | 0.46 | 0.7312 | 351.26 | 2837.63 |
| C | DMD + full $\mathcal{L}_{\text{pixel}}$ | 19.03 | 0.65 | 0.1367 | 45.94 | 718.44 |
| D | DMD + $\mathcal{L}_{\text{flow}}$ | 20.31 | 0.68 | 0.1249 | 41.27 | 398.52 |
| E | DMD + $\mathcal{L}_{\text{flow}}$ + partial $\mathcal{L}_{\text{pixel}}$ | 20.39 | 0.71 | 0.1184 | 39.72 | 363.49 |
| F | DMD + $\mathcal{L}_{\text{flow}}$ + full $\mathcal{L}_{\text{pixel}}$ | **20.43** | **0.72** | **0.1179** | **39.47** | **350.88** |

## G. More Visualizations of SpeedVFI

We present additional qualitative results of SpeedVFI in Fig. 12 and Fig. 13. These examples are drawn from our DiversityVideo dataset, which includes videos from SNU-FILM (Choi et al., 2020) and BVI-DVC (Ma et al., 2021) at a resolution of $832 \times 448$. Each sequence is interpolated to 65 frames, and we uniformly sample 8 frames for visualization. The results demonstrate that SpeedVFI produces temporally coherent and visually stable interpolations across diverse

scenarios, including driving scenes, multi-person motions, and both indoor and outdoor environments.

## H. Additional Qualitative Comparison

We provide additional qualitative comparisons against both learning-based (AMT (Li et al., 2023), BiM-VFI (Seo et al., 2025), GIMM-VFI (Guo et al., 2024)) and diffusion-based (FCVG (Zhu et al., 2025), ViBiDSampler (Yang et al., 2025a), Framer (Wang et al., 2025a)) interpolation methods, as shown in Fig. 14. Across diverse scenes, SpeedVFI consistently preserves fine-grained structures—such as limbs, foot contours, and fast-moving objects like waving swords—while maintaining temporal coherence. In contrast, optical-flow-based approaches often suffer from inaccurate motion estimation, leading to noticeable artifacts or missing details. Diffusion-based baselines may hallucinate inconsistent structures under large motion, whereas our one-step framework produces results that are both sharper and more structurally stable.

## I. Effect of Flow Matching Loss

We provide additional visualizations to further illustrate the effect of the Flow Matching loss $\mathcal{L}_{\text{flow}}$, complementing the analysis in the main paper. As shown in Fig. 11, removing $\mathcal{L}_{\text{flow}}$ leads to interpolated frames that are semantically plausible but fail to preserve identity and visual fidelity. For example, the appearance of objects such as the black swan or the car becomes noticeably distorted, despite the overall motion remaining reasonable. Since video interpolation requires faithful reconstruction of intermediate frames conditioned on the input keyframes, these results highlight the necessity of $\mathcal{L}_{\text{flow}}$ supervision for maintaining structural and appearance consistency.

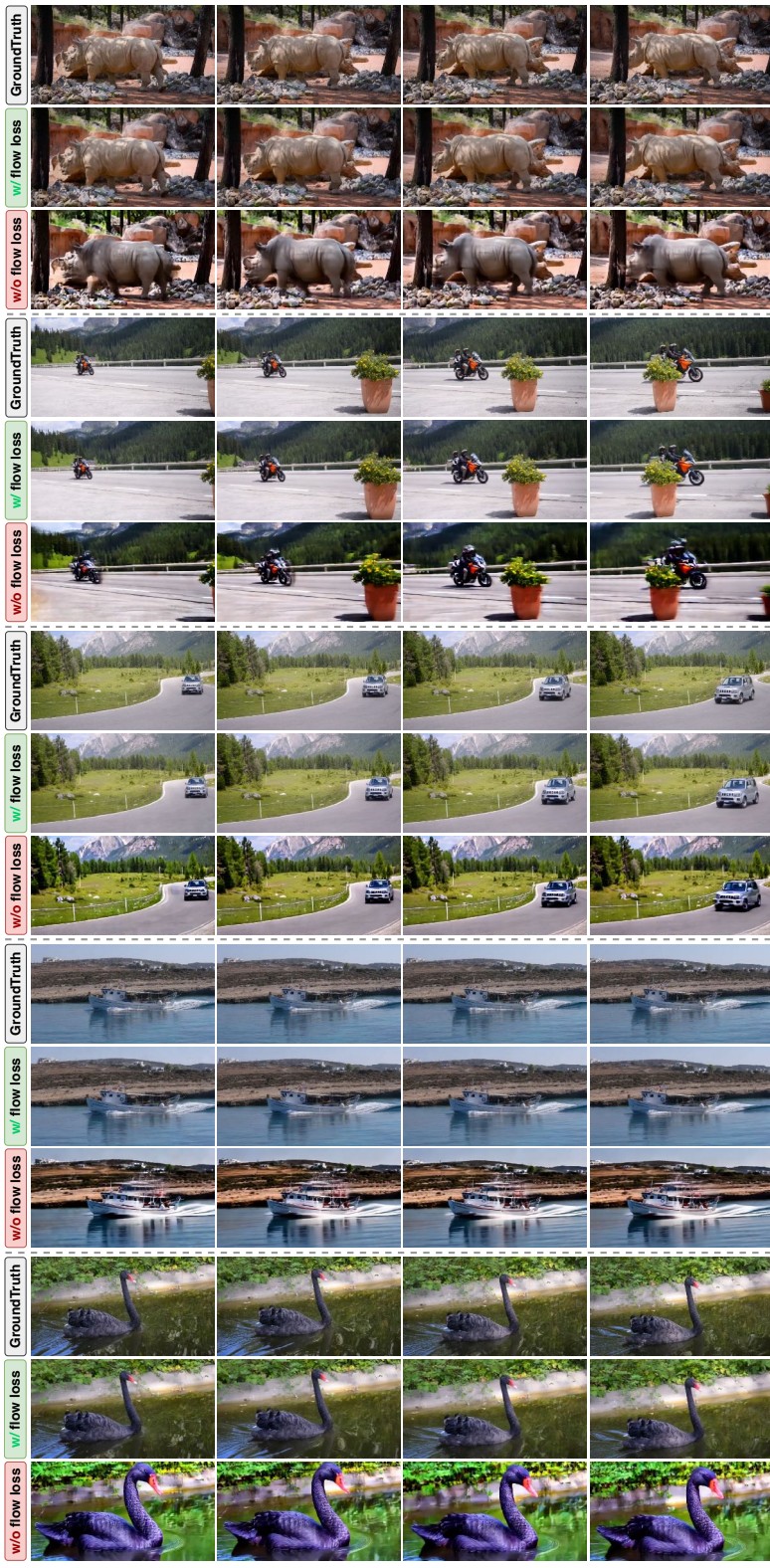

*Figure 11.* **Effect of removing the Flow Matching loss.** Without $\mathcal{L}_{\text{flow}}$, the model produces semantically reasonable but low-fidelity interpolations, often failing to preserve object identity. Incorporating $\mathcal{L}_{\text{flow}}$ yields sharper and more structurally consistent intermediate frames.

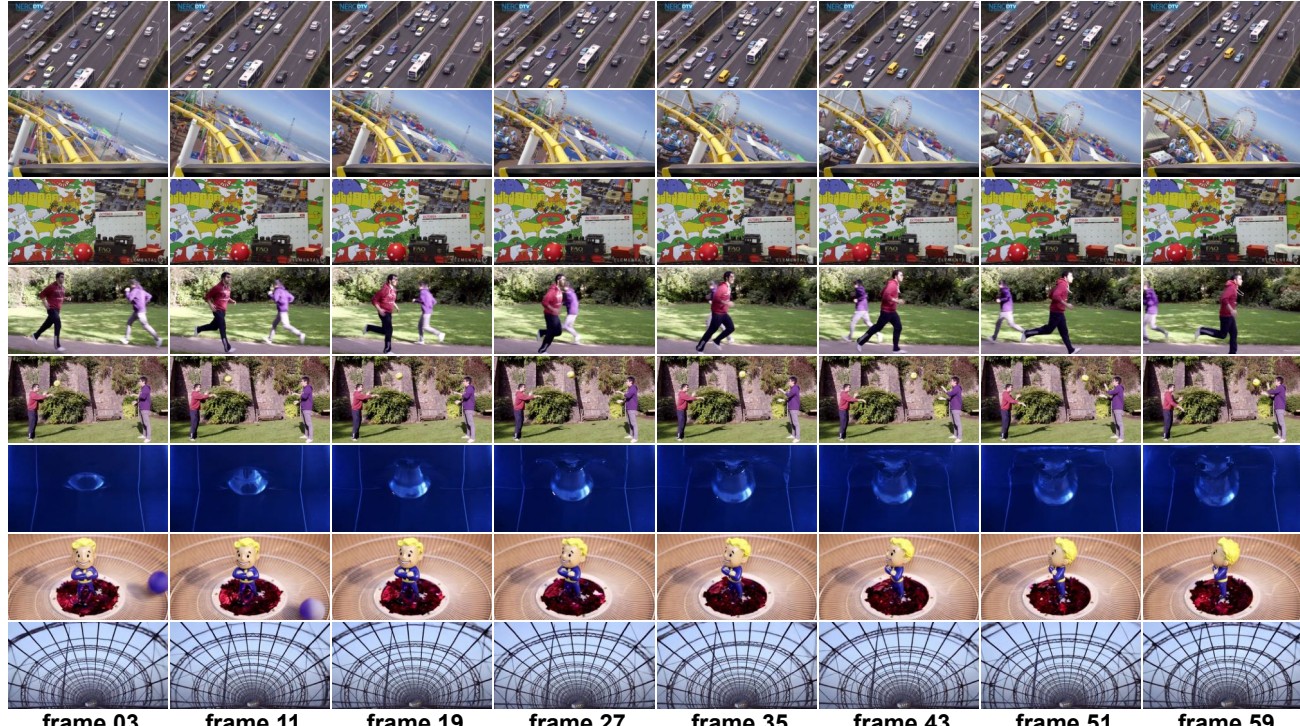

| frame 03 | frame 11 | frame 19 | frame 27 | frame 35 | frame 43 | frame 51 | frame 59 |

*Figure 12.* **Additional qualitative results of SpeedVFI on BVI-DVC (Ma et al., 2021) videos.** We show uniformly sampled frames from 65-frame interpolated sequences. SpeedVFI produces sharp structures and stable temporal transitions in challenging motion scenes.

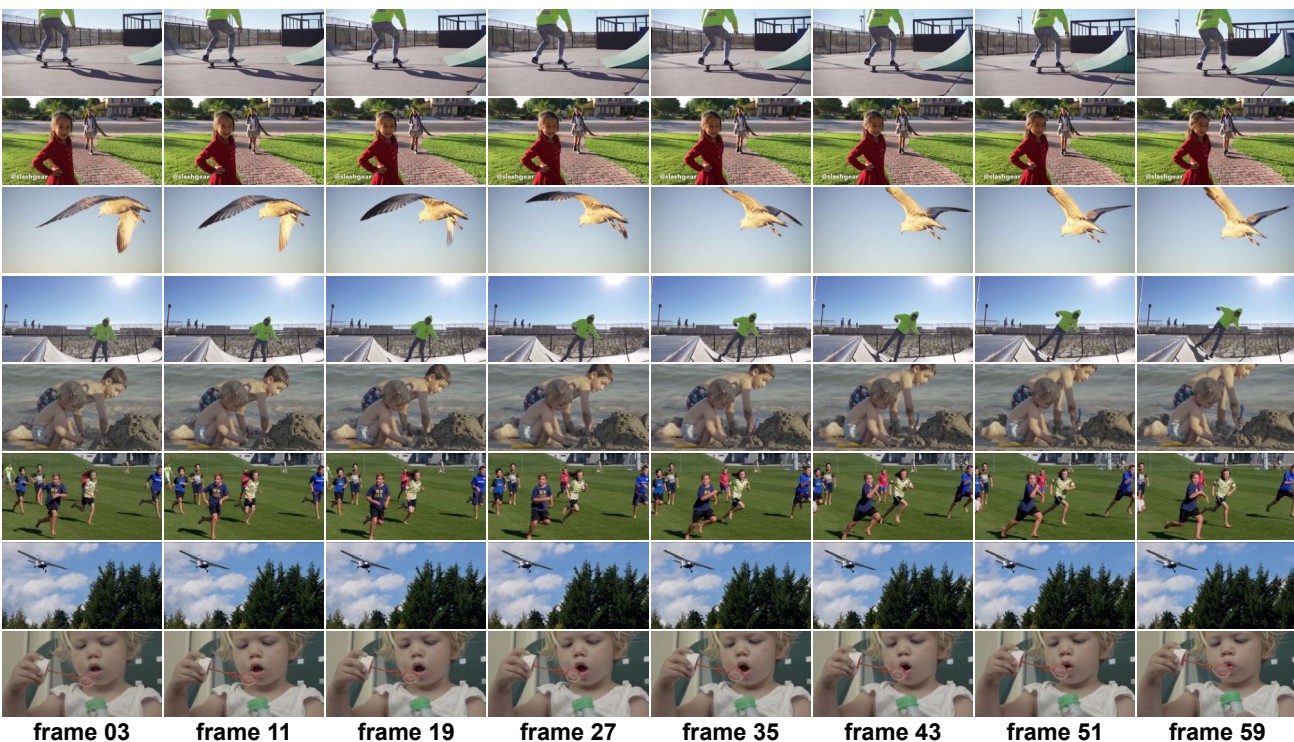

| frame 03 | frame 11 | frame 19 | frame 27 | frame 35 | frame 43 | frame 51 | frame 59 |

*Figure 13.* **Additional qualitative results of SpeedVFI on SNU-FILM (Choi et al., 2020) videos.** Our method maintains temporal coherence and visual fidelity across diverse motions and dynamic scenes.

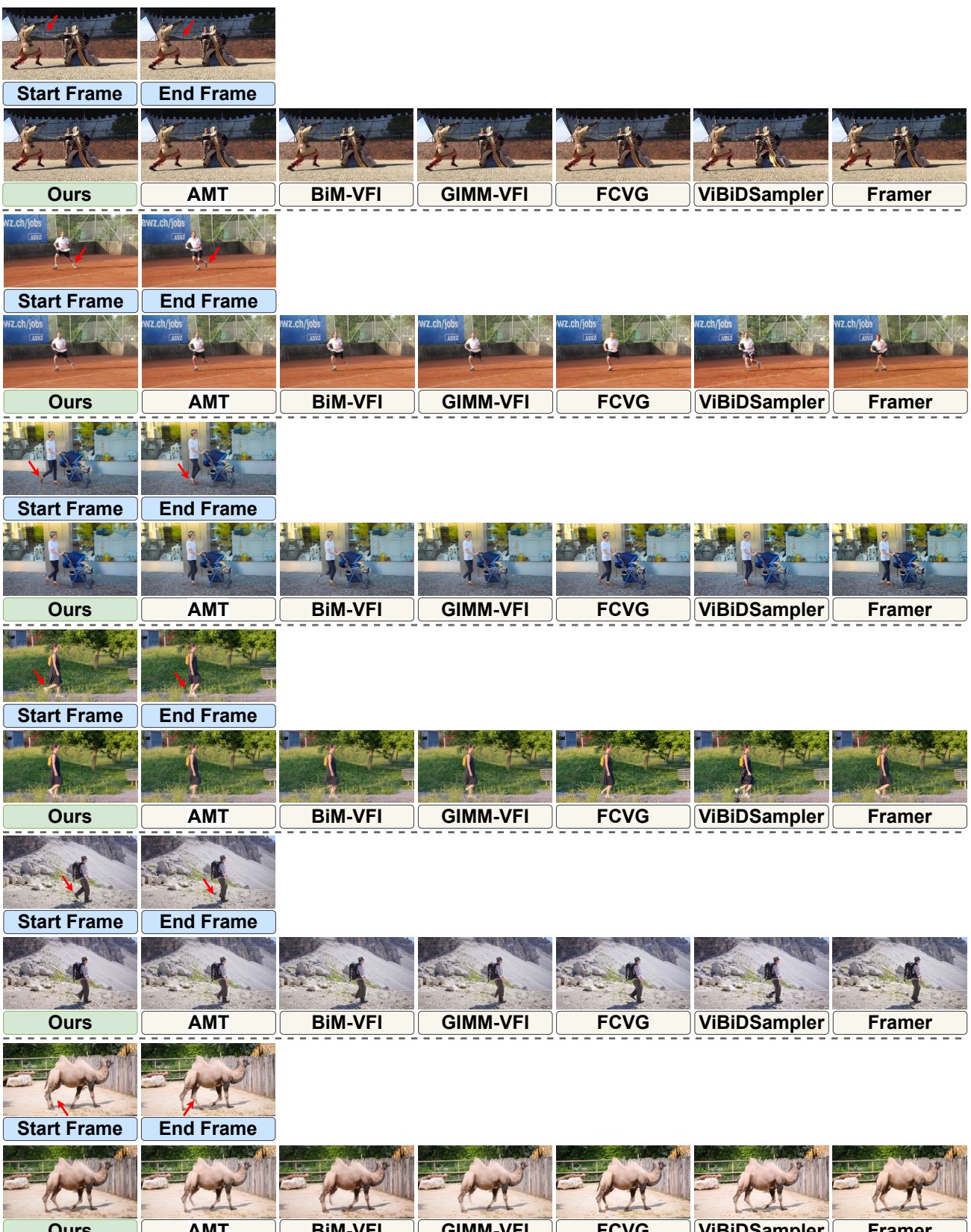

*Figure 14.* **Additional qualitative comparisons with learning-based and diffusion-based VFI methods.** SpeedVFI preserves fine-grained structures and maintains temporal coherence across challenging motions, whereas flow-based methods exhibit detail loss due to inaccurate optical-flow estimation, and diffusion-based approaches may produce inconsistent or distorted regions. Red arrows highlight areas where our method performs better.

