# OpenReview forum: "SpeedVFI: One-step Diffusion for Efficient Video Frame Interpolation"
_ICML.cc/2026/Conference — ICML 2026 regular_

### Official Review · Reviewer_v6ze · 2026-03-05

**Soundness:** 3
**Presentation:** 3
**Significance:** 3
**Originality:** 3
**Overall Recommendation:** 5
**Confidence:** 4

**Summary:**

This paper proposes SpeedVFI, a highly efficient one-step diffusion framework for video frame interpolation (VFI) that addresses the structural redundancy and iterative latency of existing multi-step, pairwise generative models. By processing the entire keyframe sequence in a single forward pass and distilling the generation trajectory into one step, the method significantly reduces computational overhead. To enable this unified architecture, the authors introduce temporal RoPE alignment to ensure positional consistency between condition and noise latents, alongside a noise-centric partial attention mechanism to reduce complexity while preserving global temporal context. Extensive experiments demonstrate that SpeedVFI achieves orders-of-magnitude acceleration over prior diffusion baselines, matching the inference speed of traditional learning-based methods without compromising visual quality.

**Compliance With Llm Reviewing Policy:**

Affirmed.

**Final Justification:**

This original and well-presented work makes a practically significant contribution by integrating one-step distillation with a unified multi-frame interpolation pipeline, achieving notable efficiency gains for diffusion-based VFI without compromising visual quality. The rebuttal resolved my concerns, albeit not perfectly. While the paper itself is a Weak Accept, I appreciate the authors' genuine effort in the rebuttal. For that, I raise my recommendation up to Accept.

**Key Questions For Authors:**

(1) Regarding the training details, the paper states that GPU memory limitations restricted the computation of the pixel loss to only the first two generated latents; how does this localized constraint impact the global structural fidelity of the entire 65-frame sequence, and have the authors considered gradient checkpointing or other memory-efficient techniques to enable sequence-wide pixel loss? If it can be shown that global fidelity remains robust or a workaround is feasible, this would directly alleviate my concerns regarding the soundness of the training methodology.
(2) While the noise-centric partial attention successfully accelerates inference, it exhibits inferior identity preservation during early training stages; does this initial instability introduce convergence risks when scaling to much longer video sequences, and how exactly does the model recover its prior? Addressing this would improve my evaluation of the framework's scalability and robustness.
(3) The current one-step distillation pipeline is exclusively initialized and trained using the Wan-TI2V-5B backbone ; have the authors tested this framework on other video diffusion architectures (like Hunyuan-Video or CogVideoX ), and does the temporal RoPE alignment transfer seamlessly? Demonstrating compatibility across different architectures would significantly strengthen the paper's claims of broad significance and originality, potentially raising my scores in those categories.

**Limitations:**

No, the authors have not adequately discussed the limitations and potential negative societal impacts of their work. Regarding societal impact, the current "Impact Statement" explicitly declines to highlight any specific consequences, which is insufficiently dismissive. The authors should revise this to thoughtfully acknowledge potential misuses of high-efficiency video generation technologies, such as the creation of deepfakes or misinformation. Furthermore, there is no dedicated discussion of technical limitations. While some constraints are briefly mentioned in the methodology—such as GPU memory restricting the pixel loss computation to only the first two latents and the initial inferior identity preservation of the partial attention mechanism —these should be consolidated into a proper limitations section. The authors should also explicitly acknowledge the limitation of validating their one-step distillation framework on only a single pretrained backbone (Wan-TI2V-5B).

**Strengths And Weaknesses:**

The paper demonstrates strong originality by uniquely combining one-step distribution matching distillation with a unified multi-frame processing pipeline to completely eliminate the structural redundancies of pairwise diffusion VFI. The proposed temporal RoPE alignment and noise-centric partial attention are clever, contextually appropriate adaptations of transformer mechanisms that successfully maintain continuous temporal geometry and reduce computational complexity. The presentation is remarkably clear, with compelling visual aids that cleanly articulate the bottlenecks of current multi-step methods versus the proposed concurrent solution. In terms of significance, achieving a 15x to 157x speedup over existing diffusion baselines while matching the inference efficiency of traditional learning-based methods represents a highly practical and substantial advancement for real-world deployment. Furthermore, the technical soundness is solidly supported by extensive quantitative benchmarking across diverse datasets and thorough ablation studies that rigorously validate the necessity of the flow matching loss and token concatenation strategies.
However, the submission exhibits a few minor weaknesses regarding experimental constraints and soundness. Due to GPU memory limitations, the pixel loss during training is computed by decoding only the first two generated latents, which might restrict the model's ability to enforce strict structural fidelity uniformly across the entire 65-frame sequence. Additionally, while the noise-centric partial attention mechanism effectively accelerates inference, the authors acknowledge it exhibits slightly inferior identity preservation and detail reconstruction during early training stages before converging, warranting further discussion on whether this poses stability risks or limits scaling to even longer sequences. Finally, the framework is exclusively initialized and trained using a single pre-trained backbone (Wan-TI2V-5B); demonstrating the adaptability of this one-step distillation framework across other distinct video diffusion architectures would further solidify the claims of broad generalizability.

---

> ### Author Rebuttal · Authors · 2026-03-31
>
> Thank you for the thoughtful review and for recognizing the originality, clarity, and practical significance of our paper. We appreciate your constructive suggestions and address them below.
>
> **1. Localized pixel loss**
>
> We agree that sequence-wide pixel supervision would be preferable if memory allowed. In our training, however, the pixel loss mainly serves as a local reconstruction/perceptual regularizer, while global structure across all generated latents is primarily enforced by the latent-space DMD and flow-matching objectives. Empirically, our 65-frame quantitative and qualitative results do not show obvious global drift despite decoding only the first two generated latents for `L_pixel`.
>
> We have already enabled gradient checkpointing for the one-step transformer, the real/fake score transformers, and the VAE, together with FlashAttention, to reduce memory usage. On a lighter `512x256`, `9 -> 33`, `4x` setting where sequence-wide decoding is feasible, adding global pixel loss further improves `20.31 / 0.68 / 0.1249 / 41.27 / 398.52` to `20.43 / 0.72 / 0.1179 / 39.47 / 350.88` (PSNR / SSIM / LPIPS / FID / FVD). We therefore view full-sequence pixel loss as beneficial but not essential for maintaining global coherence; in the main paper we prioritized the more practical high-resolution, 65-frame setting. We will clarify this trade-off and mention more memory-efficient decoding as future work.
>
> **2. Partial attention: efficiency, fidelity, and stability**
>
> Because closely related questions about partial attention were also raised by Reviewers **2hcz** and **zPcg**, we provide a consolidated clarification here. The three main takeaways are that partial attention provides a meaningful transformer-level efficiency gain, preserves fidelity after convergence, and does not show unstable optimization. Mechanistically, partial attention removes only the **condition queries**; all **noise queries** still attend to both condition and noise keys/values, so the information path for the generated frames is preserved.
>
> Under the same `832x448x65` setting, the efficiency breakdown is:
>
> | Attention | DiT denoise time (s) | End-to-end time (s) | Transformer TFLOPs |
> | --- | ---: | ---: | ---: |
> | Full | 2.41 | 10.72 | 168.54 |
> | Partial | 1.67 | 9.98 | 127.66 |
>
> This corresponds to a `1.44x` speedup in the denoising transformer and a `24.25%` reduction in transformer FLOPs. The end-to-end gain is smaller because Tab. 2 in the main paper also includes VAE encode/decode, which already takes about `1.64s` and `6.0s` in the same setting. Therefore, the system-level gain is diluted by the large fixed cost outside the attention blocks, while the transformer-side gain is substantial.
>
> For fidelity under DAVIS `9 -> 33`, `4x`:
>
> | Attn | Step | PSNR | SSIM | LPIPS | FID | FVD |
> | --- | ---: | ---: | ---: | ---: | ---: | ---: |
> | Full | 1K | 20.11 | 0.65 | 0.1377 | 41.42 | 370.77 |
> | Partial | 1K | 20.06 | 0.63 | 0.1480 | 43.76 | 382.57 |
> | Partial | 2K | **20.14** | **0.68** | **0.1212** | **37.68** | **347.53** |
>
> When compared after the same `1K training steps`, partial attention is slightly weaker than full attention, matching our qualitative observation of a small early optimization lag. With longer optimization (`2K` steps), partial attention surpasses the `1K` full-attention baseline across all reported metrics. This suggests that the gap is mainly an optimization-speed issue rather than a convergence risk or an intrinsic loss of representational capacity.
>
> **3. Single-backbone validation**
>
> We agree that validation on `Wan-TI2V-5B` alone is a limitation of the current submission. We chose Wan because its stronger VAE spatial compression (16x) offers a favorable balance between video generation quality and token efficiency, which makes long-range one-step distillation more practical. More importantly, our proposed components are architecture-level: the one-step distillation pipeline is not tied to Wan specifically, and temporal RoPE alignment is likely to transfer to other latent video diffusion transformers that use RoPE-style temporal embeddings. We have not yet completed a systematic cross-backbone study on models such as Hunyuan-Video or CogVideoX, we will present this as future work.
>
> **4. Limitations and societal impact**
>
> We agree that the current Impact Statement is too brief. In the revision, we will add a dedicated limitations/impact discussion covering localized pixel-loss supervision, memory/computation growth with sequence length, validation on a single backbone, and the possible misuse of efficient video generation for harmful content such as deepfakes or misinformation.

---

> > ### Author Rebuttal · Reviewer_v6ze · 2026-04-01
> >
> > Thank you for your reply. It has alleviated some of my concerns. However, since the ablation suggests that adding $L_{\text{pixel}}$ yields measurable improvements, I am not yet convinced that whether it is merely beneficial or essential. Besides, whether temporal RoPE alignment would transfer to architectures remains a reasonable hypothesis rather than empirical evidence. I'll maintain my current score.

---

> > > ### Author Response · Authors · 2026-04-07
> > >
> > > Thank you for the helpful follow-up. We agree that our earlier wording around $L_{pixel}$ should have been more precise. To directly address your concern, we ran two additional experiments.
> > >
> > > **1. Is $L_{pixel}$ merely beneficial, or actually important?**
> > >
> > > We compared six variants under the same setting (`512x256, 9 -> 33, 4x, 1K` training steps), where GPU memory is sufficient for full-sequence decoding and thus *full* $L_{pixel}$ supervision.
> > >
> > > **Note**:
> > > - *partial* $L_{pixel}$: apply pixel loss on the first two generated latents
> > > - *full* $L_{pixel}$: apply pixel loss on all generated latents
> > >
> > > | Variant | Pixel supervision | PSNR↑ | SSIM↑ | LPIPS↓ | FID↓ | FVD↓ |
> > > | --- | --- | ---: | ---: | ---: | ---: | ---: |
> > > | A |  $L_{DMD}$ | 12.54 | 0.12 | 0.8217 | 402.60 | 4184.76 |
> > > | B |  $L_{DMD}$ + *partial* $L_{pixel}$ | 16.37 | 0.46 | 0.7312 | 351.26 | 2837.63 |
> > > | C |  $L_{DMD}$ + *full* $L_{pixel}$ | 19.03 | 0.65 | 0.1367 | 45.94 | 718.44 |
> > > | D |  $L_{DMD}$ + $L_{flow}$ | 20.31 | 0.68 | 0.1249 | 41.27 | 398.52 |
> > > | E |  $L_{DMD}$ + $L_{flow}$ + *partial* $L_{pixel}$ | 20.39 | 0.71 | 0.1184 | 39.72 | 363.49 |
> > > | F |  $L_{DMD}$ + $L_{flow}$ + *full* $L_{pixel}$ | **20.43** | **0.72** | **0.1179** | **39.47** | **350.88** |
> > >
> > > These six variants show two distinct regimes. Under **DMD-only** training (A/B/C), pixel supervision is indeed very important: adding *partial* $L_{pixel}$ already helps substantially, and *full* $L_{pixel}$ improves the results much further. However, once **flow-matching supervision** is added (D/E/F), the model already remains strong and globally coherent even without pixel loss, and $L_{pixel}$ becomes an additional fidelity booster rather than a strict prerequisite: `D < E < F`, but the gaps are much smaller than in the DMD-only case.
> > >
> > > Therefore, $L_{pixel}$ is clearly beneficial and should not be downplayed, but it is also **not the sole or indispensable mechanism for maintaining global coherence in the final recipe**. The main global structure is already established by DMD + flow supervision, while partial pixel supervision recovers most of the remaining fidelity gain at much lower decoding cost than full-sequence decoding. Full $L_{pixel}$ is still the strongest setting when it is feasible, but these results also support the reasonableness of our main-paper choice to supervise only the first two generated latents under limited memory budget. In this sense, $L_{pixel}$ is helpful, but not strictly necessary for maintaining global coherence in the final setting.
> > >
> > > **2. Does temporal RoPE alignment transfer beyond Wan?**
> > >
> > > We agree that our earlier statement was only a reasonable architectural hypothesis without direct evidence. To address this point, we additionally ran a small-scale validation on a second RoPE-based video diffusion backbone, `cosmos-predict-2.5-2B`, under the same `832x448, 9 -> 33, 4x, 1K-step` setting:
> > >
> > > | Methods | PSNR↑ | SSIM↑ | LPIPS↓ | FID↓ | FVD↓ |
> > > | --- | ---: | ---: | ---: | ---: | ---: |
> > > | Interleaved | 17.87 | 0.57 | 0.2081 | 92.33 | 946.83 |
> > > | Sequential  | 18.31 | 0.61 | 0.1968 | 85.49 | 810.06 |
> > > | Sequential + temporal RoPE alignment | **18.56** | **0.63** | **0.1780** | **68.12** | **720.05** |
> > >
> > > These results suggest that temporal RoPE alignment is not specific to Wan2.2-TI2V-5B. On a second RoPE-based video diffusion backbone, it again improves over both interleaved and naive sequential token concatenation.
> > >
> > > We also sincerely appreciate your concerns about the role of $L_{pixel}$ and the generality of the model design. These are important questions, and they helped us sharpen both the experiments and the claims. We hope the additional evidence above further clarifies these points and helps address your remaining concerns.

---

### Official Review · Reviewer_zPcg · 2026-03-10

**Soundness:** 2
**Presentation:** 2
**Significance:** 2
**Originality:** 2
**Overall Recommendation:** 4
**Confidence:** 3

**Summary:**

This paper aims to address the fundamental problem of iterative latency in diffusion-based VFI framework, and the pairwise redundancy of processing the same input frames multiple times. To alleviate the problem of iterative latency, the authors propose a one-step distillation pipeline based on DMD, for one-step acceleration, along with tailored flow matching loss and pixel loss. For pairwise redundancy, the paper study on how to position the condition and target (noisy) latents, and introduce temporal RoPE alignment, which is basically normalizing the timestamps of condition and target latents. To further save redundant computational cost, the paper propose noise-centric partial attention, where only the queries of noise latents participate in attention operations. The proposed pipline shows superior latency and performance compared to existing studies.

**Compliance With Llm Reviewing Policy:**

Affirmed.

**Final Justification:**

My main concern of the initial paper was that it did not include discussions on computational costs, even though the paper was on designing a lightweight, accelerated framework. However during the rebuttal period, the authors have quantitatively shown that their proposed method could save great amount of computational cost, along with cost-performance tradeoff figure, and have resolved my main concern. Including the discussions on computational costs from the rebuttal period in the final version would be necessary.

**Key Questions For Authors:**

1. In line 238-239, left column, the authors argue that interleaving two latent sequences in time breaks temporal continuity of noise trajectory; however, I did not clearly understand this part. I wonder what the authors meant in failure of "temporal continuity", as I do not think the naïve solution does not break it.
2. Could the authors further provide explanations on the flow matching loss? Theoretically speaking, how does this loss help in "motion" continuity (right column, lines 160-161)? The ablation experiment in Fig.9 of the Appendix show that the flow matching loss helps in preserving object identity, but the explanation in the main paper sort of contradicts this, and the provided explanation also does not persuasive in the current form. Additionally, the flow loss $\mathcal{L}_\text{flow}$ is a loss function defined without any variables, but is sort of used as a functional form with variables in Eq.4. Stricter notation would be helpful.

**Limitations:**

No; I could not find limitations or potential negative impacts mentioned. The method could possibly be used in harmful content generation.

**Strengths And Weaknesses:**

Strength:
- The problem definition is clear, and the proposed method adequately solves the mentioned problems.
- Experimental results with better latency and performance seem to be strong.


Weaknesses:
- Experiments
    - Efficiency: the paper is mainly on improving the efficiency of VFI methods. Although the time and NFE is reported, I think it would be better by explicitly reporting the number of FLOPs of the proposed framework for fair comparison. Furthermore, I suggest comparison against RIFE[1] and MoMo[2], as they are strong studies on improved efficiency and performance, although they are not based on pre-trained models. MoMo in specific, is another diffusion-based approach, so I think it would be important to see a comparison. Many conventional VFI methods like these (described as learning-based in the paper), are relatively lightweight compared to recent pre-trained diffusion-based methods. Therefore, I think a cost-performance tradeoff comparison would be very helpful, as this paper mainly aims for improved efficiency. (Does not necessarily have to be a form of tradeoff graph, but at least simple comparisons both on performance and computational cost.)
   - Ablation: this is not a problem simply for this paper, but I am not sure if the comparison with the diffusion-based baselines can be considered to be fair, as each method adopt different pre-trained diffusion models. The performance gap could have came from the backbone pre-trained model, with stronger models, with efficiently designed architectures. The provided ablation study does show that the proposed method is effective, but I am not sure if the comparison itself with prior studies have been conducted strictly fairly. In addition, in the experiment of full attention vs partial attention, the paper shows improved efficiency (latency) and qualitative results, but did not show quantitative results in performance. I wonder how the performance could be maintained with partial attention.
    - Missing citation: this paper is based on Rotary Positional Embedding (RoPE) [3], but is never cited in the paper.

---

> ### Author Rebuttal · Authors · 2026-03-31
>
> Thank you for the constructive review and for recognizing the clear problem definition and strong latency/quality results. We address the main points below.
>
> **1. Comparison with RIFE/MoMo**
>
> We additionally compare with RIFE [1] and MoMo [2]; the quantitative and qualitative results are provided in Tab. 1 and Fig. 1 of [this supplementary file](https://anonymous.4open.science/api/repo/icml2026_rebuttal-1B45/file/For_reviewer_zPcg.pdf?v=b3feedf0). Although MoMo uses diffusion, we group it as a flow-based method because it predicts optical flow and then synthesizes frames through flow-based warping, rather than directly generating frames as we do. A key limitation of flow-based synthesis is that it cannot hallucinate newly exposed content in occluded regions, and its performance also degrades when flow estimation becomes unreliable under fast or complex motion. As shown in Fig. 1, both MoMo and RIFE fail to recover structures such as the camel's legs and building pillars.
>
> **2. FLOPs and cost-performance trade-off**
>
> We also report FLOPs. Under the same `832x448`, `17 -> 65`, `4x`, one-step setting, SpeedVFI uses `660.2 TFLOPs` in total (`127.66` transformer, `85.01` VAE encode, `447.43` VAE decode); partial attention reduces transformer cost from `168.54` to `127.66` (`24.25%`). We also provide a cost-performance comparison in terms of PSNR, inference time, and FLOPs; please see Fig. 2 in [this file](https://anonymous.4open.science/api/repo/icml2026_rebuttal-1B45/file/For_reviewer_zPcg.pdf?v=b3feedf0).
>
> **3. Comparison fairness**
>
> We agree that cross-paper comparisons across different pretrained backbones are not fully controlled. To reduce this confound, we additionally reproduced `ViBiDSampler` with the `Wan2.2-TI2V-5B` backbone (`ViBiDSampler-Wan`); even then, SpeedVFI remains much cheaper (`660.2` vs. `38210 TFLOPs`) and more accurate (`27.12` vs. `23.0` PSNR). Within-backbone ablations also support our claim: `L_DMD` alone gives `12.60 / 1627.51` (PSNR / FVD), adding VFI-specific flow and pixel constraints improves this to `20.05 / 417.85`, and RoPE alignment improves over interleaved/plain sequential concatenation (`FVD 641.64 / 518.06` vs. `417.85`). Thus, the gains cannot be explained by backbone choice alone.
>
> **4. Partial attention: quantitative fidelity and efficiency**
>
> We agree that the quantitative fidelity of partial attention should be stated more explicitly. Since closely related concerns were also raised by Reviewers 2hcz and v6ze, we summarize the key takeaway here. *Please refer to Reviewer v6ze, Point 2 for the full efficiency/fidelity tables and additional details.*
>
> Partial attention reduces transformer cost from `168.54` to `127.66 TFLOPs` and denoising-transformer time from `2.41s` to `1.67s`, while fidelity is only slightly below `Full@1K` when compared after the same `1K training steps` and recovers with longer optimization (`Partial@2K`: `20.14 / 0.68 / 0.1212 / 37.68 / 347.53` vs. `Full@1K`: `20.11 / 0.65 / 0.1377 / 41.42 / 370.77`, PSNR / SSIM / LPIPS / FID / FVD). So we view this as a small optimization lag, not a fundamental weakness of the design.
>
> **5. What we mean by "temporal continuity"**
>
> The reason interleaving hurts `temporal continuity` is that the unknown/noise latents no longer form a continuous trajectory in token order: adjacent unknown frames become separated by condition tokens, so local temporal interactions among generated frames are interrupted and the token spacing no longer matches their physical timestamps. Our aligned RoPE preserves the physical positions of both known and unknown frames without breaking the noise trajectory.
>
> **6. Flow matching loss and Eq. 4**
>
> We agree that the current explanation and notation can be clearer. The key term validated in our ablation is $L_{flow}(v_{pred}, v_{gt})$: it directly matches the predicted latent trajectory to the ground-truth latent trajectory. This helps motion smoothness because it reduces abrupt changes in the latent dynamics across adjacent frames, and it also helps identity consistency because a trajectory that stays closer to the ground-truth evolution reduces frame-to-frame appearance drift.
>
> The second term, $L_{flow}(v_{pred}, v_{fake})$, is an auxiliary regularizer used within our DMD-based training to keep the predicted trajectory aligned with the fake trajectory used during distillation. It is not the main term studied in the ablation, and we agree that the current notation makes this role insufficiently clear. In the revision, we will rewrite Eq. 4 to distinguish the two terms more clearly.
>
>
> **7. Citation and limitations**
>
> Thank you for pointing this out. We will cite the original RoPE paper in the revision. We also agree that the paper should discuss limitations and potential negative impacts more clearly, including the current single-backbone evaluation setting and the possible misuse of efficient video generation for harmful content such as deepfakes or misinformation.

---

> > ### Author Rebuttal · Reviewer_zPcg · 2026-04-02
> >
> > I thank the authors for the clarifications. Most of the explanations provided have resolved my questions. I strongly suggest the discussions on computational costs (FLOPs) to be included in the final version.
> >
> > Also, I would like to ask for clarification on any possible misunderstandings I have. The time-performance-cost bubble graph visualization was very intuitive and helpful, but I found it a little misleading. Specifically, the bubble size of the proposed method is not clearly visualized in the attached figure. While the bubble size (circle) resemble the computational cost (FLOPs), the bubble of SpeedVFI (Ours) is a star-shape, and not a bubble, so it is hard to interpret the actual computational cost. Then I checked to see the actual number of FLOPs, which said that SpeedVFI requires about 660 TFLOPs and ViBiDSampler to take 38210 TFLOPs. I also tried looking up the actual numbers of FLOPs, and found that MoMo was reported to spend about 1 TFLOPs. Despite the 38210x difference, the bubble figure does not seem to appropriately show visualize this difference. Could the authors clarify on this?

---

> > > ### Author Response · Authors · 2026-04-03
> > >
> > > Thank you for the helpful follow-up. We agree that the earlier bubble plot was not ideal for reading exact FLOPs, especially because our method was marked with a star and the FLOPs span several orders of magnitude. To make this clearer, we now provide (1) a pure numeric table below and (2) a revised visualization with consistent circular markers. Please also refer to the updated figure here: [link](https://anonymous.4open.science/api/repo/icml2026_rebuttal-1B45/file/For_reviewer_zPcg2.pdf?v=6d96ae7d).
> > >
> > > We also noticed a plotting error in the earlier bubble figure: the PSNR of ViBiDSampler (Wan) was mistakenly shown as `23.08` instead of `20.08`. The updated figure corrects this value.
> > >
> > >
> > > **1. Pure numeric comparison**
> > >
> > > All numbers below are reported under the same `17 -> 65`, `4x`, `832x448` evaluation setting. FLOPs denote **total inference FLOPs for the full task**, not per-pair or per-step FLOPs.
> > >
> > > | Method | PSNR | Inference time (s) | FLOPs (TFLOPs) |
> > > | --- | ---: | ---: | ---: |
> > > | *Learning-based* |  |  |  |
> > > | RIFE | 23.75 | 4.20 | 4.54 |
> > > | MoMo | 24.14 | 10.20 | 76.87 |
> > > | AMT-S | 23.56 | 10.32 | 4.11 |
> > > | AMT-L | 23.64 | 10.56 | 21.93 |
> > > | BiM-VFI | 26.90 | 14.16 | 43.35 |
> > > | GIMM-VFI | 27.10 | 10.54 | 277.36 |
> > > | *Diffusion-based* |  |  |  |
> > > | ViBiDSampler (SVD) | 19.89 | 371.40 | 17584.26 |
> > > | ViBiDSampler (Wan) | 20.08 | 139.83 | 38213.20 |
> > > | FCVG | 22.30 | 1570.18 | 13637.28 |
> > > | Framer | 22.79 | 150.69 | 10773.10 |
> > > | Ours | 27.12 | 9.98 | 660.19 |
> > >
> > > For pairwise methods such as RIFE and MoMo, the reported FLOPs are accumulated over all forward passes needed to complete the full `17 -> 65`, `4x` interpolation task.
> > >
> > > **2. Why MoMo is `76.87 TFLOPs` here, while its paper reports about `1 TFLOP`**
> > >
> > > To clarify, we re-measured MoMo under the same `2x` setting used in our accounting. This gives `1.6014 TFLOPs` for one pairwise `2x` pass of the full `MoMo.forward()` pipeline.
> > >
> > > The smaller number reported in the MoMo paper comes from a different scope: it reports about `1.12 TFLOPs` for one `2x` pass of the motion diffusion U-Net itself, while our `1.6014 TFLOPs` includes both the diffusion U-Net and the final frame synthesis model.
> > >
> > > For the full `17 -> 65`, `4x` task used in our table, MoMo must be applied recursively over `16` adjacent intervals, with `3` pairwise `2x` forwards per interval, i.e., `48` total forward passes. This gives a total of `1.6014 x 48 = 76.87 TFLOPs`.
> > >
> > > So the two numbers are not contradictory: the paper reports a single-pass, U-Net-only count, while our table reports total task-level FLOPs for the complete inference pipeline under a common setting. We will make this accounting scope explicit in the final version to avoid confusion.
> > >
> > >
> > > **3. Conclusion**
> > >
> > > To avoid ambiguity in how bubble size is interpreted, we now show both a raw-area version, where bubble area is directly proportional to total FLOPs, and a sqrt-compressed version, which is easier to read in the low/mid-cost regime.
> > >
> > > The revised plots are intended as task-level comparisons under the same `17 -> 65`, `4x`, `832x448` setting. Under this view, SpeedVFI is clearly more efficient than prior diffusion-based VFI methods, with higher PSNR, lower latency, and much lower total FLOPs. Compared with learning-based methods, SpeedVFI achieves the highest PSNR and competitive inference time, although its FLOPs are still higher. To make this trade-off easier to read, we now show both a raw-area version and a sqrt-compressed version, and we will include this revised bubble figure and the accompanying FLOPs discussion in the final version.
> > >
> > > We hope these clarifications resolve your concerns and make the contribution clearer.

---

### Official Review · Reviewer_MjCd · 2026-03-12

**Soundness:** 2
**Presentation:** 3
**Significance:** 2
**Originality:** 3
**Overall Recommendation:** 3
**Confidence:** 3

**Summary:**

To address the low inference efficiency of diffusion-based video frame interpolation methods, the paper proposes SpeedVFI, a one-step diffusion framework. This method performs interpolation across the entire video sequence through a single forward pass, and combines knowledge distillation to compress the generation process into a single step. In addition, temporal RoPE alignment and a noise-centric partial attention mechanism are introduced to preserve temporal consistency. The experimental results suggest that the proposed method improves computational efficiency; however, the performance improvement over existing learning-based methods is not clearly demonstrated.

**Compliance With Llm Reviewing Policy:**

Affirmed.

**Final Justification:**

While the proposed framework is reasonably motivated and clearly presented, my concerns remain only partially addressed after the rebuttal. On the one hand, in several examples (e.g., Fig. 6, rows 1 and 4), the proposed method does not outperform existing approaches such as Framer. Although the rebuttal attributes this to a sharpness–smoothness trade-off, this explanation lacks systematic empirical support. In particular, it has not been validated through targeted analysis on challenging scenarios such as small-scale objects or fast and complex motions, where such effects are most likely to occur. As a result, the adequacy of this explanation remains unclear.
On the other hand, the contribution of the method mainly lies in the integration of existing techniques, with relatively limited novelty at the methodological level. In summary, although the work has certain value, its level of novelty and empirical support are not yet sufficiently convincing.

**Key Questions For Authors:**

1. Although the proposed global joint interpolation strategy replaces pairwise processing through unified modeling and shows advantages in structural consistency and computational efficiency, its potential impact on optimization difficulty and modeling flexibility still requires further discussion and analysis.

2. The method generates all intermediate frames simultaneously in a single forward pass, which improves temporal consistency but also introduces strong coupling across frame predictions. It remains unclear whether this design limits the model’s ability to correct local interpolation errors, particularly in challenging scenarios such as severe occlusions, fast non-rigid motion, or large viewpoint changes.


3. The current runtime evaluation is mainly conducted under a fixed setting (65 frames at 832×448 resolution). It would be beneficial to further analyze latency under different resolutions, sequence lengths, and interpolation ratios to provide a more comprehensive assessment of the method’s efficiency.

**Limitations:**

It would be helpful for the authors to further analyze the robustness boundaries of the proposed method in challenging scenarios (e.g., severe occlusions or complex motion) to better clarify its applicability.

**Strengths And Weaknesses:**

Strengths：
The paper addresses two key inefficiencies in diffusion-based video frame interpolation, namely the redundant computation introduced by pairwise interpolation and the inference latency caused by multi-step denoising. To this end, the authors propose a one-step diffusion framework, SpeedVFI, which jointly processes the entire keyframe sequence in a single forward pass. This design simultaneously removes the redundancy of pairwise processing and the time cost of iterative sampling. Overall, the proposed method partially narrows the efficiency gap between diffusion-based and learning-based approaches, but the performance improvement remains limited. The manuscript is generally well organized and clearly written. However, existing learning-based VFI methods appear to have already achieved a good balance between efficiency and reconstruction quality. Although the paper attempts to improve the inference efficiency of diffusion-based approaches, it does not sufficiently demonstrate their distinctive advantages over learning-based methods.

Weaknesses：
Although the method improves inference efficiency, all intermediate frames are generated simultaneously within a single forward pass, resulting in highly coupled frame predictions. This design may limit the model’s ability to independently correct local interpolation errors, and therefore requires further analysis and discussion.

---

> ### Author Rebuttal · Authors · 2026-03-31
>
> Thank you for the constructive review and for recognizing that our paper addresses the key inefficiencies of diffusion-based VFI and is clearly written. We agree the paper should explain more clearly what is gained, and what remains limited.
>
> **1. Why generative VFI remains useful beyond learning-based methods**
>
> We agree this point should be stated more directly. Our claim is not that generative VFI already beats strong learning-based methods across the board. The point is that interpolation is often ambiguous in cases like large motion, heavy occlusion, or newly exposed regions, where local matching or deterministic regression can miss structures. In those cases, generative VFI is still useful because it can synthesize plausible missing content instead of only warping what is already visible.
>
> From this perspective, the contribution of SpeedVFI is to make generative VFI much more practical. We keep the benefit of a generative formulation, but remove much of the cost through single-pass multi-frame interpolation and one-step denoising. So our message is not that learning-based VFI is no longer needed, but that generative VFI still has a clear role in harder, more ambiguous cases, and SpeedVFI makes that trade-off much more favorable.
>
> **2. Optimization difficulty and modeling flexibility**
>
> Joint generation does not collapse all missing frames into one rigid global code. Each target frame still has its own noise latent, aligned temporal coordinate, and query in the transformer. The model therefore keeps frame-wise flexibility, while allowing each unknown frame to use global temporal context from the full keyframe sequence. Also, because all target frames are predicted in parallel from separate noise latents, there is no autoregressive-style error accumulation. We will clarify in the revision that the coupling in SpeedVFI is intended to improve information sharing and temporal consistency, not to remove local controllability.
>
> **3. Does joint generation limit local correction?**
>
> In severe occlusion, fast non-rigid motion, and large viewpoint changes, a local pair often does not contain enough evidence. Giving each target frame access to the full keyframe sequence can therefore help disambiguate local structure rather than restrict it. This is consistent with our qualitative results on thin structures under occlusion (flamingo legs, Fig. 6, third row; the man's lower leg, Fig. 10, third row from bottom; camel legs, Fig. 10, bottom row), fast non-rigid motion (running horse, Fig. 4, fourth row), large displacement / viewpoint change (motorcycle, Fig. 6, first row), and complex motion (fast-running child, Fig. 11, third row from bottom; running horse, Fig. 4, fourth row). In these cases, pairwise or flow-based methods often lose structure while our jointly conditioned generator preserves more complete details.
>
>
> **4. Runtime beyond the fixed 65-frame, 832x448 setting**
>
> We agree that broader runtime analysis is useful. To address your question directly, we additionally profiled latency under different resolutions, output lengths, and interpolation ratios on a single NVIDIA A800. For easier reference, we also provide an anonymous supplementary table in LaTeX format here: [anonymous supplementary table](https://anonymous.4open.science/api/repo/icml2026_rebuttal-1B45/file/For_reviewer_MjCd.pdf?v=8d3d1cfc)
>
> | Resolution | Output frames | 8x cond. frames | 8x time (s) | 4x cond. frames | 4x time (s) |
> | --- | ---: | ---: | ---: | ---: | ---: |
> | 416x224 | 33 | 5 | 4.31 | 9 | 4.59 |
> | 416x224 | 65 | 9 | 5.75 | 17 | 5.92 |
> | 832x448 | 33 | 5 | 7.14 | 9 | 7.39 |
> | 832x448 | 65 | 9 | 10.47 | 17 | 10.63 |
>
> These results show the expected trend: latency increases with resolution and output length, while `8x` is slightly faster than `4x` for the same output length because it uses fewer condition frames. We will add this trend more clearly in the revision.

---

> > ### Author Rebuttal · Reviewer_MjCd · 2026-04-04
> >
> > Thank you for the authors’ response. I generally agree that generative VFI holds an advantage over learning-based methods in absolutely ambiguous scenarios, where the interpolated frames contain content with no corresponding visual information in the preceding and subsequent keyframes. However, from the results presented in the original paper—for instance, the first row of Figure 6— it can be observed that in object motion scenarios, the proposed method does not demonstrate advantage over the Framer method (Wang et al., 2025a); instead, it blurs the originally static and clear scenes. Therefore, while the design of generating all intermediate frames simultaneously in a single forward pass offers certain benefits, it also leads to strong coupling between frame predictions. Particularly in challenging scenarios such as severe occlusions, fast non-rigid motion, or large viewpoint changes, it remains unclear whether this design limits the model’s ability to correct local interpolation errors. The additional runtime analysis is appreciated. However, I notice an inconsistency in the hardware settings: the main paper reports results on an NVIDIA A100, while the supplementary analysis is conducted on an NVIDIA A800.

---

> > > ### Author Response · Authors · 2026-04-07
> > >
> > > Thank you for the valuable follow-up. We agree that this is a reasonable concern for a joint-interpolation paradigm, and we appreciate you highlighting this potential risk. We also agree that Fig. 6, row 1 does not show an advantage over Framer in that particular fast-motion scene. However, we do not think this single frame is sufficient evidence that joint generation introduces harmful frame coupling.
> > >
> > > We believe this case more likely reflects a sharpness-smoothness trade-off of the one-step generator under very fast motion. If harmful coupling were the dominant issue, we would expect more systematic and temporally persistent local failures, especially on small objects, thin structures, dense textures, or motion-sensitive regions. The current qualitative evidence does not show such a pattern. Fig. 4 in the main paper, together with Figs. 10 and 11 in the supplementary material, already shows stable motion across sequences such as the running horse, the approaching bus, and the boatman striking the water. Furthermore, in Fig. 6, row 2, our method correctly preserves the small ball in the lower-right region, while Framer misses it entirely. In Fig. 6, row 3, our method restores the flamingo's left leg, a very thin structure that many methods fail to recover; Framer keeps the leg visible, but its position is less consistent with the motion, since the left leg should move to overlap with the right leg.
> > >
> > > To make the temporal behavior more explicit, **we additionally provide consecutive-frame visualizations here**: [high-resolution PDF](https://anonymous.4open.science/api/repo/icml2026_rebuttal-1B45/file/For_reviewer_MjCd2.pdf?v=b368756d) and [compressed PDF](https://anonymous.4open.science/api/repo/icml2026_rebuttal-1B45/file/For_reviewer_MjCd2_compressed.pdf?v=e72bfeeb). The two files contain the same content; the compressed version may be easier to load if the high-resolution file is slow to open.
> > >
> > > These sequences show that even when Framer appears sharper in an isolated frame, its motion can be less temporally coherent overall. In the bubble-blowing sequence (a), the mouth should remain open throughout, but Framer predicts it as closed in several middle frames. In the throwing/catching-ball sequence (b), Framer misses the ball in some middle frames and the trajectory becomes jerky, including brief backward motion, whereas our ball motion remains continuous. In the pedestrian-running sequence (d), Framer produces visible artifacts in multiple frames and the human structure becomes unstable, while our result remains more stable. We hope these supplementary visualizations make the temporal difference clearer.
> > >
> > > Taken together, the current evidence does not suggest that joint interpolation introduces a harmful coupling effect that systematically limits local correction. Instead, SpeedVFI remains competitive or advantageous when small structures or under-determined content need to be recovered from broader temporal context. More broadly, the practical advantage of SpeedVFI is its overall quality-efficiency trade-off: under the common `832x448, 17 -> 65, 4x` setting, Framer requires `150.69s` and `10773.10 TFLOPs`, while SpeedVFI requires `9.98s` and `660.19 TFLOPs`, with competitive or better overall quantitative performance in the main table. This is the main motivation of our design, and we will revise the final version to make this scope more precise.
> > >
> > > Regarding runtime, the main paper reports the original benchmark on an NVIDIA A100, while the additional experiments during the rebuttal period were run on an NVIDIA A800 because the original A100 server was temporarily unavailable. We used the A800 results only to analyze relative scaling trends across resolutions, sequence lengths, and interpolation ratios. These measurements are consistent with the main runtime table and do not conflict with it (e.g., the main-paper `832x448, 17 -> 65, 4x` result is `9.98s`, while the rebuttal measurement is about `10.63s`). To avoid any ambiguity, we will rerun this supplementary table on the original A100 server once it becomes available and report unified A100 results in the final version.
> > >
> > > We sincerely thank you for your careful review. We appreciate both your concern about potential frame coupling and your careful observation of the change in our experimental setting. We hope that the additional analysis and supplementary visualizations help address these concerns, and we believe your comments will directly help us make the final version more complete and more reliable.

---

### Official Review · Reviewer_2hcz · 2026-03-12

**Soundness:** 3
**Presentation:** 3
**Significance:** 2
**Originality:** 2
**Overall Recommendation:** 3
**Confidence:** 4

**Summary:**

This paper introduces a new video frame interpolation model, SpeedVFI. The proposed model uses a single forward pass to avoid redundant keyframe duplication and applies temporal RoPE to better align noise latents with global conditions. SpeedVFI trains a one-step diffusion model via a DMD pipeline and substitutes full-attention blocks with their partial counterparts to improve efficiency. Experiments demonstrate that SpeedVFI substantially accelerates interpolation while maintaining competitive performance with SOTA methods.

**Compliance With Llm Reviewing Policy:**

Affirmed.

**Final Justification:**

The paper is clearly presented, with a framework that effectively improves efficiency while maintaining competitive performance. However, the originality is limited, as the method primarily integrates existing techniques, and the evaluation is insufficient to justify the design choices fully. The rebuttal clarified several aspects and partially addressed my concerns, but key issues regarding ablations and validation remain. Overall, I maintain my original score.

**Key Questions For Authors:**

1. Quantitative comparisons between full- and partial-attention designs are suggested to better demonstrate the trade-offs between efficiency and fidelity. Given the limited efficiency gains from partial attention, I question the appropriateness of this design.
2. How does GPU memory usage scale with video length in the proposed framework, and could memory requirements limit its applicability to longer videos?
3. Why is DMD chosen as the distillation strategy for this task, and how does it compare with alternative distillation approaches (e.g., consistency distillation or progressive distillation) in this setting?

**Limitations:**

yes

**Strengths And Weaknesses:**

### Strengths

1. The paper is clearly written and well structured, with a logical presentation of the problem, method, and empirical evaluation.
2. The work identifies and addresses the structural redundancy introduced by pairwise processing in diffusion-based VFI pipelines.
3. The proposed framework substantially reduces inference cost while maintaining competitive interpolation quality compared with existing methods.



### Weaknesses

1. The proposed method largely builds on existing techniques, including DMD, flow matching, and RoPE, and its primary contribution lies in their integration for efficient VFI, resulting in limited methodological novelty.
2. The ablation study on partial attention is limited. Quantitative comparisons and qualitative evaluations in challenging scenarios are lacking, and the potential trade-off between efficiency and fidelity is not fully explored.
3. The evaluation is conducted primarily under a fixed configuration (17 input keyframes with 4× interpolation), which limits the generalization validation.
4. The analysis of the distillation process is insufficient. The training dynamics and stability of the DMD-based distillation are not thoroughly examined, and the rationale for adopting DMD is not clearly explained or justified.

---

> ### Author Rebuttal · Authors · 2026-03-31
>
> Thank you for the thoughtful review and for recognizing the paper's clarity, the importance of the redundancy issue, and the efficiency gains of SpeedVFI.
>
> **1. Methodological novelty**
>
> We agree that our paper does not propose a completely new diffusion component by itself. Rather, the main novelty is a new **formulation for efficient diffusion-based VFI** that jointly removes the two dominant bottlenecks in prior generative VFI: repeated pairwise processing across intervals and iterative denoising within each interval. Concretely, SpeedVFI combines single-pass interpolation over the entire keyframe sequence with one-step distilled denoising in a unified VFI pipeline.
>
> Importantly, this is not a plug-and-play combination of existing components. In our setting, `$L_{DMD}$` alone is insufficient for interpolation quality: it gives only `12.60 PSNR`, while adding our VFI-specific flow and pixel constraints improves the result to `20.11 PSNR`. Likewise, naive interleaving or plain sequential concatenation are inferior to temporal RoPE alignment (`FVD 641.64 / 518.06` vs. `417.85`). In addition, our noise-centric partial attention further reduces cost while maintaining fidelity after convergence. We will revise the wording to make this contribution claim more precise.
>
> **2. Partial attention: efficiency and fidelity**
>
> We agree that this point should be clarified more carefully. Since closely related questions were also raised by Reviewers zPcg and v6ze, we summarize the key result here. *Please refer to Reviewer v6ze, Point 2 for the full efficiency/fidelity tables and additional details.*
>
> Under `832x448x65`, partial attention reduces denoising-transformer time from `2.41s` to `1.67s` (`1.44x`) and transformer FLOPs from `168.54` to `127.66` (`24.25%` reduction). When compared at the same `1K training steps`, the fidelity gap is small, and it disappears with longer optimization (`Partial@2K`: `20.14 / 0.68 / 0.1212 / 37.68 / 347.53` vs. `Full@1K`: `20.11 / 0.65 / 0.1377 / 41.42 / 370.77`, PSNR / SSIM / LPIPS / FID / FVD). So we view this as a mild early optimization-speed trade-off, not evidence that partial attention fundamentally hurts fidelity.
>
> **3. Fixed configuration and generalization**
>
> We agree that broader benchmarking across sequence lengths and interpolation ratios would strengthen the paper; this is a valid limitation. We chose the `17 -> 65`, `4x` setting as our main evaluation protocol because `4x` interpolation is a commonly used VFI setting, especially for frame-rate upsampling and slow-motion generation, and 65 output frames already contain `16` interpolation intervals, which clearly expose pairwise redundancy and cumulative denoising latency.
>
> Importantly, the method itself is not tied to this specific protocol. Our ablations were first conducted under a different `9 -> 33`, `4x` setting to determine the most effective design choices, and the final `17 -> 65` model was then trained based on that validated configuration. This already suggests that the framework is not hard-coded to a single sequence length, although broader multi-setting evaluation would further strengthen the paper and will be stated as a limitation.
>
> **4. GPU memory scaling with video length**
>
> We measured peak GPU memory at fixed resolution `832x448` and fixed `4x` interpolation while varying the video length:
>
> | Conditioning keyframes | Generated frames | Peak memory (GiB) |
> | ---: | ---: | ---: |
> | 5 | 17 | 20.40 |
> | 9 | 33 | 20.48 |
> | 13 | 49 | 20.54 |
> | 17 | 65 | 20.62 |
> | 21 | 81 | 20.68 |
> | 25 | 97 | 20.75 |
>
> From `17` to `97` generated frames, peak memory increases by only about `0.35 GiB` (`~1.7%`) in our tested setup. Within this range, memory growth is modest and does not appear to be the main bottleneck.
>
> **5. Why DMD for one-step distillation, and how stable is it?**
>
> We chose DMD2 because our target is **true one-step generation** for VFI. Progressive distillation typically compresses the sampler gradually and often remains multi-step unless multiple distillation stages are applied. Consistency-based methods are also relevant one/few-step alternatives. In addition, the original DMD2 paper reports strong one-step generation quality in its benchmark settings, making it a relatively mature and practical starting point for our setting.
>
> We don't think DMD is the only possible distillation strategy for VFI. Rather, we view it as a suitable one-step backbone that still requires substantial task-specific adaptation. The evidence in our paper is that $L_{DMD}$ alone is insufficient, $L_{flow}$ substantially improves temporal coherence and identity preservation, and TTUR further improves the distilled model (`interval=5` is better than `interval=1` on LPIPS/FID/FVD). In other words, DMD alone is not enough for VFI; it becomes effective in our setting only after the proposed VFI-specific constraints and training strategy are added.

---

> > ### Author Rebuttal · Reviewer_2hcz · 2026-04-03
> >
> > Thank you for the detailed rebuttal. The authors clarified several design choices (e.g., partial attention and DMD) and provided additional analysis.
> >
> > However, I still have concerns that the evaluation (e.g., limited settings and ablations) is not yet sufficient to justify the design choices and generalization fully.

---

> > > ### Author Response · Authors · 2026-04-08
> > >
> > > Thank you for the follow-up and for clearly summarizing your remaining concerns in the final justification. We also appreciate your positive assessment of the paper's clarity, and that our earlier rebuttal had already clarified part of the design rationale. We regret that, due to limited time and compute during the rebuttal period, we were not able to complete the new experiments early enough for your earlier acknowledgement. We therefore understand that you maintained your current score. Since the new results below directly address the remaining concerns about ablations and broader validation, we hope you may still find them useful and take them into account in your final assessment.
> > >
> > > > NOTE: These results were measured on an A800 because the A100 server used in the main paper was temporarily unavailable.
> > >
> > > **1. Cross-setting and challenging-case validation of partial attention**
> > >
> > > In addition to the main-paper ablation at `832x448, 9 -> 33, 4x`, we evaluated partial attention under two additional settings: `416x224, 9 -> 33, 4x` and `832x448, 17 -> 65, 4x`, so that both resolution and sequence length are varied.
> > >
> > > Attention|Train steps|PSNR↑|LPIPS↓|FVD↓|DiT time (s)↓|End-to-end time (s)↓|DiT TFLOPs↓
> > > -|-:|-:|-:|-:|-:|-:|-:
> > > `416x224, 9 -> 33, 4x`
> > > Full   |`1K`|20.25|0.1192|390.11|1.19|4.94|16.34
> > > Partial|`1K`|20.20|0.1257|410.84|0.90|4.59|10.93
> > > Partial|`2K`|**20.48**|**0.1153**|**309.10**|0.90|4.59|10.93
> > > `832x448, 9 -> 33, 4x`
> > > Full   |`1K`|20.11|0.1377 |370.77|2.12|8.08|75.44
> > > Partial|`1K`|20.06|0.1480|382.57|1.44|7.39|53.80
> > > Partial|`2K`|**20.14**|**0.1212**|**347.53**|1.44|7.39|53.80
> > > `832x448, 17 -> 65, 4x`
> > > Full|`1K`|18.14|0.4104|921.80|2.63|11.48|168.54
> > > Partial|`1K`|18.02|0.4523|1002.7|1.78|10.63|127.66
> > > Partial|`2K`|**18.15**|**0.3492**|**890.31**|1.78|10.63|127.66
> > >
> > > Across all three settings, partial attention consistently reduces transformer-side cost (`DiT time` / `DiT TFLOPs`) while the fidelity gap at the same training step remains small. With longer optimization, this gap is recovered and the `Partial@2K` variant slightly surpasses the `Full@1K` baseline in all three settings. This makes the efficiency-fidelity trade-off more convincing than in our earlier response, and suggests that the benefit of partial attention is not tied to a single protocol.
> > >
> > > **2. Additional evaluation under a different interpolation ratio**
> > >
> > > To further evaluate whether SpeedVFI generalizes beyond the main `17 -> 65, 4x` setting, we additionally fine-tuned our model to the `832x448, 9 -> 65, 8x` setting and compared it with representative baselines on the DiversityVideo benchmark used in the main paper:
> > >
> > > Method|PSNR↑|SSIM↑|LPIPS↓|FID↓|FVD↓|Inference time (s)↓|End-to-end TFLOPs↓
> > > -|-:|-:|-:|-:|-:|-:|-:
> > > *Learning-based*
> > > RIFE|28.04|0.91|0.1813|38.90|355.71|5.28|5.3027
> > > MoMo|29.83|0.91|0.1714|33.69|286.46|12.55|89.68
> > > *Diffusion-based*
> > > ViBiDSampler (Wan)|19.79|0.69|0.3920|116.54|2769.87|86.93|26759.93
> > > Framer|23.15|0.77|0.2398|34.53|814.63|92.72|5386.56
> > > Ours|**31.62**|**0.93**|**0.1146**|**27.88**|**204.14**|10.47|629.75
> > >
> > > Under this harder `8x` protocol, SpeedVFI again shows strong overall quality and remains much more efficient than the diffusion-based baselines. We also provide qualitative comparisons for this setting in [the supplementary PDF](https://anonymous.4open.science/api/repo/icml2026_rebuttal-1B45/file/For_reviewer_2hcz.pdf?v=fe85a128), including challenging cases with tiny-object motion and non-rigid human motion. This shows that the overall behavior of SpeedVFI is not restricted to the original `17 -> 65, 4x` setting.
> > >
> > > **3. Novelty**
> > >
> > > Since originality is also mentioned in the final justification, we briefly clarify that point here as well. DMD, flow matching, and RoPE are already widely used in recent generative models. Our goal is not to claim novelty in these primitives themselves, but to make diffusion-based VFI genuinely efficient. In our view, the contribution lies in three aspects: **(1) problem formulation**: reformulating generative VFI from repeated pairwise interpolation into unified one-step sequence interpolation to remove structural redundancy; **(2) task-specific adaptation**: adapting these components through temporal RoPE alignment, noise-centric partial attention, and a DMD-based recipe with additional flow and pixel supervision; and **(3) system-level efficiency**: eliminating pairwise duplication and iterative denoising latency in generative VFI while preserving strong interpolation quality. For these reasons, **we believe SpeedVFI is better characterized as a task-specific reformulation and adaptation of generative VFI, rather than a simple aggregation of existing techniques.**
> > >
> > > Your suggestions have also been very helpful in strengthening the paper. We sincerely appreciate the time and care you have taken in reviewing the paper, and we hope these new results help further address the remaining concerns behind your final justification.

---

### Decision · Program_Chairs · 2026-04-30

**Decision:**

Accept (regular)

**Comment:**

This paper presents a one-step diffusion framework for VFI. By processing the entire video sequence in one forward pass, the proposed method is more efficient than competing methods for VFI.

Reviewers liked the good empirical results and well-motivated design. They also shared concerns about missing ablation studies, computational cost analysis, and unclear motivation for some technical choices.

After the rebuttal and discussion phase, two reviewers recommended acceptance and two reviewers recommended rejection.

On the positive side, the rebuttal resolved reviewer zPcg's concerns about missing comparisons to RIFE and MoMo. The reply to reviewer v6ze also demonstrated the value of the pixel loss and the generalizability of the RoPE alignment beyond the WAN model.

On the other hand, reviewers 2hcz and MjCd maintained their weak rejection ratings after discussion, both citing limited technical novelty. Reviewer MjCd also pointed out that under certain scenarios (small objects, large motion), it is unclear how the proposed method compares with existing approaches such as Framer.

The AC read all the reviews and discussion. The proposed method clearly demonstrates a new design in VFI with good empirical results on improving efficiency. Although the influence of some design choices is still unclear, for example whether joint prediction of all intermediate frames would lead to strongly coupled frames and how it compares with existing methods for small objects under large motion, it can serve as a good baseline for the community to study this new design.

The decision is to recommend the paper for acceptance. The authors are encouraged to revise the paper to incorporate the comments from the rebuttal period in the final version.